

# On unifying carbonate rheology

James Gilgannon[1] and Marco Herwegh[2]

[1]School of Geographical and Earth Sciences, University of Glasgow, Glasgow, United Kingdom
[2]Institute of Geological Sciences, University of Bern, 3012 Bern, Switzerland

**Correspondence:** James Gilgannon (james.gilgannon@glasgow.ac.uk)

**Abstract.** We review the results from twenty three experimental works conducted on the rheology of carbonates from the last fifty years to revisit the long-noted discordance in the experimental results from a range of limestones and marbles. Such a exercise is needed to bring together the various datasets generated in the twenty three years since the last major review, as many of them observe relationships that run contrary to existing rheological models. By revisiting the large data set, we find that most low and high stress experimental measurements can be explained by the combined effect of grain size and the molar fraction of magnesium carbonate ($XMgCO_3$). Our results highlight that much of the calcite-dolomite series exists in a continuum of strength that changes with $XMgCO_3$. In contrast to previous findings, we establish that diffusion creep in calcite is sensitive to both grain size and magnesium content, showing that an increase in $XMgCO_3$ acts to weaken a rock. While in dislocation creep we confirm the observation that $XMgCO_3$ has a strengthening effect but extend it beyond synthetic Mg-calcite samples to natural starting materials . Most notably our results suggest that when the composition of a carbonate is factored in then grain size can be shown to have a weakening effect in dislocation creep for fine grained rocks. This is the opposite finding to the currently accepted flow law for calcite rocks in the dislocation creep regime where a decrease in grain size strengths a rock. We contextualise these new results by combining them with data from natural shear zones to show that carbonates are much weaker than would be expected from previous flow laws in a crustal section. Ultimately our review provides new pragmatic flow laws for carbonates in the calcite-dolomite series for diffusion and dislocation creep.

## 1 Introduction

During mountain building, marine sediments are incorporated into an orogeny and often become subsequently important tectonic horizons (e.g. Pfiffner, 1982; Burkhard, 1990; Bestmann et al., 2000; Herwegh and Pfiffner, 2005; Ebert et al., 2007; Nania et al., 2022). Many of these sediments are calcareous and they have a range of compositions, distributions of mineral phases and grain sizes inherited from their depositional environments. Importantly these chemical and micro-structural characteristics can change with time as the material experiences metamorphism and deformation (Ebert et al., 2007). Figure 1 shows this variety, both in composition, in this case equilibrated molar fraction of magnesium ($XMgCO_3$) and deformationally induced steady state grain size. Understanding the effect these changes have on rock rheology is of great importance for eval-





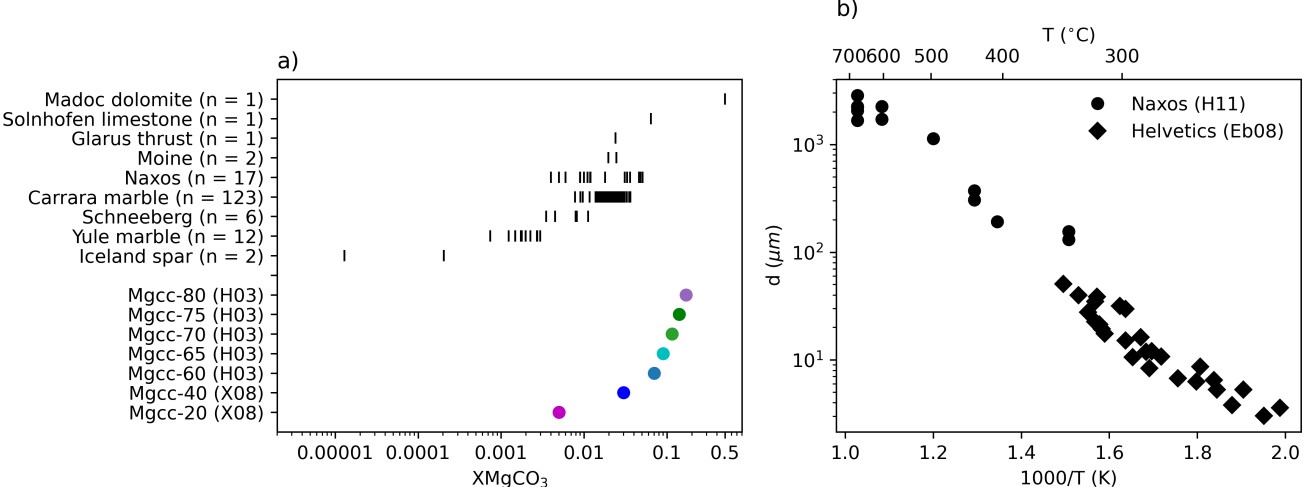

**Figure 1.** Compilation of measurements from natural and synthetic carbonates. Figure 1a shows the variation in the amount of magnesium, while figure 1b shows the how the steady state grain sizes of carbonate mylonites change with temperature.

uating crustal deformation and is most readily established through laboratory deformation experiments.

Over the years experiments on carbonates have explored the role of different lattice impurities (Herwegh et al., 2003; Freund et al., 2004; Xu et al., 2008; Davis et al., 2008; Holyoke et al., 2013), the distribution of minor secondary phases (Olgaard, 1978; Walker et al., 1990; Austin et al., 2014), the effect of incorporating proportions of a stronger phase (Bruhn et al., 1999; Rybacki et al., 2003; Renner et al., 2007; Delle Piane et al., 2009; Kushnir et al., 2015), water content (Rutter, 1972; Olgaard, 1978; de Bresser et al., 2005) and the role of other microstructural characteristics like grain-size (Schmid et al., 1977; Walker et al., 1990; Rutter, 1995; Pieri et al., 2001; Renner et al., 2002; Herwegh et al., 2003; Barnhoorn et al., 2004; Xu et al., 2008). All of these variables have been shown to have some effect on rock strength, which makes understanding the deformation behaviour of calcite rich rocks complex.

Here we revisit and review a range of data from deformation experiments preformed on several carbonates with a focus on the role of grain size and magnesium content. Our motivation for doing so is that, since the last major review on calcite rheology (Renner and Evans, 2002), several new experiments have been conducted that further illuminated the role of these two important variables (e.g. Pieri et al., 2001; Herwegh et al., 2003; Barnhoorn et al., 2004; Xu et al., 2008). In particular we 40 are interested in the fact that some of the more recent experiments may not conform to the constitutive relation that is currently used to unify dislocation creep in carbonates, the modified Peierls law (Renner et al., 2002; Renner and Evans, 2002). For example, Barnhoorn et al. (2004) found that during shear zone formation and grain size reduction through dynamic recrystallisation lead to material weakening and not strengthening as would be expected by the modified Peierls law. So far, these newer



results and older data have not been compiled and comprehensively compared to one another. Making an up to date comparison

is of particular interest for carbonates because of the long standing observation of a lack of conformity between the high stress rheologies of various calcite single crystals, limestones and marbles (cf. Renner et al., 2002; Renner and Evans, 2002).

In this contribution we collate data from a range of deformation experiments and make this needed comparison. Starting from a core set of deformation experiments that record both grain size and magnesium content, trends are revealed and then

compared to a wide range of data spanning the whole magnesium carbonate series to ultimately provide a perspective and flow laws that unify observations from most carbonate deformation experiments to date.

## 2  Problems with the current model

A primary insight of Renner and Evans (2002) was that experiments gained strength with decreasing grain size (fig. 2a).

However that observation was made in the absence of chemistry. If experimental data which also had chemical characterisation reported are plotted to visualise the variation of magnesium (fig. 2b), then a different trend emerges - material strength increases with increasing magnesium. This fits with the experimental observations of Xu et al. (2008) where synthetic carbonates in the dislocation creep field were shown to harden with the concentration of magnesium. Continuing to account for chemistry also reveals that high stress domains of experimental data associated with the dislocation creep field are in fact grain size weakening

(fig. 2c), and not hardening as predicted by the modified Peierls law used by both Renner et al. (2002) and Xu et al. (2008) to fit data. All of these observations highlight that while both grain size and magnesium are clearly important for carbonate rheology, we have not fully understood their effects. The following methods, results and discussion attempt to better characterise the effect of grain size and magnesium across a range of *nominally pure* and second-phase rich magnesium carbonates.

## 3  Methods

We revisit data from 23 publications that conducted carbonate deformation experiments as well as several studies that performed chemical and microstructural characterisations of naturally deformed carbonates. Data were collated from both reported values in tables and extracted from figures of publications. The rest of this section outlines how data were converted to the appropriate units for comparison.



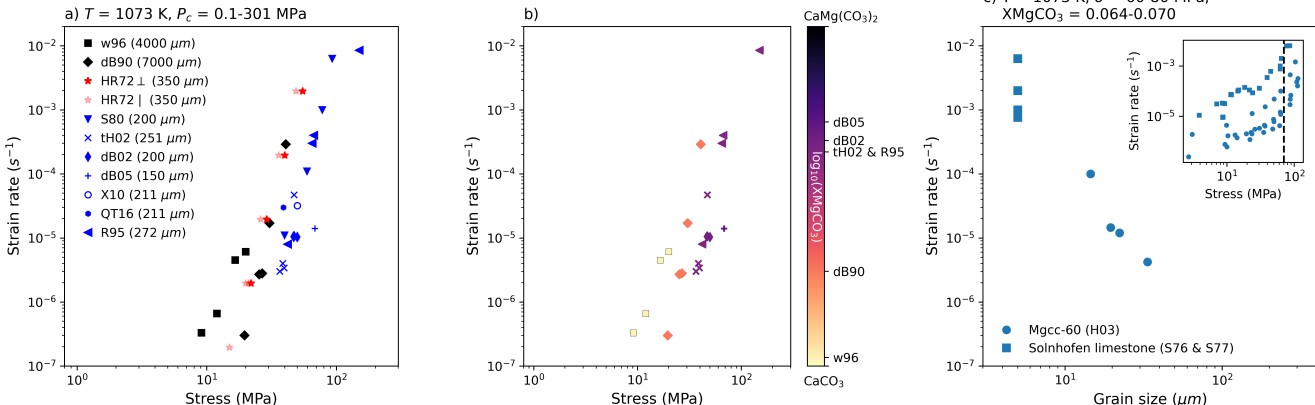

**Figure 2.** A starting point for evaluating carbonate rheology. Figure 2a visualises the core observation for the current understanding of calcite rheology, rocks are stronger for smaller grain sizes (cf. Renner and Evans, 2002). Figure 2b shows an alternative trend where a rock strengthens with magnesium. Figure 2c highlights a consequence of this observation, that grain size actually has a weakening effect if data are observed at a fixed magnesium content.

Table 1: Reported and converted magnesium values of naturally deformed carbonates and experimental starting material.

| Material | Reported value | | | | XMgCO$_3$ |
|---|---|---|---|---|---|
| | $\omega_{Mg}(ppm)$ | $\omega_{MgCO_3}$ | MgO (wt%) | XMgCO$_3$ | |
| **Moine** | | | | | |
| Oden (1985) (n = 2) | | | | 0.0197[a] | |
| | | | | 0.0245 [b] | |
| **Schneeberg** | | | | | |
| Schenk et al. (2005) (n = 6) | | | 0.14[a] | | 0.003474 |
| | | | 0.45[b] | | 0.011154 |
| **Naxos** | | | | | |
| Schenk et al. (2005) (n = 5) | 1209.1[a] | | | | 0.004974 |
| | | | 0.48[b] | | 0.011896 |
| Covey-Crump and Rutter (1989) (n = 12) | | | | 0.004[a] | |
| | | | | 0.051[b] | |

[a] Minimum value

[b] Maximum value

[c] Minimum from range from Sample D2-W (2k-7), both inside and outside of shear zone

[d] Maximum from range from Sample D2-W (2k-7), both inside and outside of shear zone





| Material | Reported value | | | | XMgCO$_3$ |
|---|---|---|---|---|---|
| | $\omega_{Mg}(ppm)$ | $\omega_{MgCO_3}$ | MgO (wt%) | XMgCO$_3$ | |
| **Synthetic calcite** | | | | | |
| Walker et al. (1990) | | | | 0.000 | |
| Renner et al. (2002) | | | | 0.000 | |
| **Synthetic Mg-calcite** | | | | | |
| Xu et al. (2008) | | | | 0.005[a] | |
| | | | | 0.030[b] | |
| Herwegh et al. (2003) | | | | 0.070[a] | |
| | | | | 0.170[b] | |
| **Iceland Spar** | | | | | |
| Wang et al. (1996) (n = 2) | 3.2[a] | | | | 0.000013 |
| | 10.5[b] | | | | 0.000043 |
| de Bresser and Spiers (1990) | 49.8 | | | | 0.000205 |
| **Yule marble** | | | | | |
| McGee (1999) (n = 10) | | | 0.03[a] | | 0.000744 |
| | | 0.0025[b] | | | 0.002966 |
| Busenberg and Plummer (1983) | | | | 0.002740 | |
| Busenberg and Plummer (1989) | | | | 0.002700 | |
| **Carrara marble** | | | | | |
| Rutter (1995) | 2250 | | | | 0.009250 |
| Covey-Crump (1998) | 2250 | | | | 0.009250 |
| Pieri et al. (2001) | | | 0.73[a] | | 0.018074 |
| | | | 0.99[b] | | 0.024487 |
| Barnhoorn et al. (2005) | | | 0.30 | | 0.007440 |
| Ter Heege et al. (2002) | 2250 | | | | 0.009250 |
| de Bresser (2002) | 3321 | | | | 0.013234 |
| de Bresser et al. (2005) | 4804 | | | | 0.019718 |
| | 5845 | | | | 0.023974 |

[a] Minimum value

[b] Maximum value

[c] Minimum from range from Sample D2-W (2k-7), both inside and outside of shear zone

[d] Maximum from range from Sample D2-W (2k-7), both inside and outside of shear zone



| Material | Reported value | | | | XMgCO$_3$ |
|---|---|---|---|---|---|
| | $\omega_{Mg}(ppm)$ | $\omega_{MgCO_3}$ | MgO (wt%) | XMgCO$_3$ | |
| Oesterling (2004) (n = 43) | | | | 0.013761[c] | |
| | | | | 0.028560[d] | |
| Solnhofen limestone | | | | | |
| Barnhoorn et al. (2005) (n=1) | | | 0.22[a] | | 0.054160 |
| | | | 0.30[b] | | 0.073626 |

[a] Minimum value

[b] Maximum value

[c] Minimum from range from Sample D2-W (2k-7), both inside and outside of shear zone

[d] Maximum from range from Sample D2-W (2k-7), both inside and outside of shear zone

Table 2: Experimentally deformed carbonates

| Experiment | | [1]$T$ [K] | [2]$P_c$ [MPa] | [3]d (d$_p$) [$\mu m$] | [4]$f_p$ [$\mu m$] |
|---|---|---|---|---|---|
| *Synthetic calcite* | | | | | |
| Walker et al. (1990) | (w90) - | | | | |
| | C50 R | 973 | 200 | 7.50 | −− |
| | C69 R | 973 | 200 | 3.40 | −− |
| Renner et al. (2002) | (R02) - | | | | |
| | C192 | 1077 | 300 | 8.00 | −− |
| | C189 | 1076 | 300 | 11.00 | −− |
| | C376 | 1078 | 300 | 20.00 | −− |
| *Synthetic Mg-calcite* | | | | | |
| Xu et al. (2008) | (X08) - | | | | |
| *Mgcc-20* | | | | | |
| | C948 | 1073 | 300 | 27.00/19.00 | −− |
| *Mgcc-40* | | | | | |
| | C942 | 1058 | 300 | 25.00/20.00 | −− |

[1] T = deformation temperature

[2] $P_c$ = confining pressure

[3] d/d$_p$ = grain-size/second phase grain-size

[4] $f_p$ = area/volume fraction of second phases

−− not applicable/not reported

* value taken from Barnhoorn et al. (2005)



| Experiment | | $^1T$ [K] | $^2P_c$ [MPa] | $^3$d (d$_p$) [$\mu m$] | $^4f_p$ [$\mu m$] |
|---|---|---|---|---|---|
| Herwegh et al. (2003) | (H03) - | | | | |
| *Mgcc-60* | | | | | |
| | cd-77 | 1073 | 300 | 14.55 | –– |
| | cd-58 & 85 | 973 | 300 | 19.50 | –– |
| | | 1073 | 300 | 19.50 | –– |
| | cd-38 | 1073 | 300 | 19.50 | –– |
| | cd-53 | 1073 | 300 | 22.23 | –– |
| | cd-50 | 1073 | 300 | 33.48 | –– |
| *Mgcc-65* | | | | | |
| | cd-61 | 973 | 300 | 17.44 | –– |
| | cd-61 & 46 | 1073 | 300 | 17.44 | –– |
| *Mgcc-70* | | | | | |
| | cd-80 | 973 | 300 | 15.01 | –– |
| | cd-80 & 56 | 1073 | 300 | 15.01 | –– |
| | cd-62 | 1073 | 300 | 17.56 | –– |
| | cd-51 | 1073 | 300 | 22.69 | –– |
| *Mgcc-75* | | | | | |
| | cd-81 & 41 | 1073 | 300 | 13.54 | –– |
| *Mgcc-80* | | | | | |
| | cd-37 | 1073 | 300 | 11.91 | –– |
| | cd-45 | 1073 | 300 | 14.89 | –– |
| | cd-49 | 1073 | 300 | 17.34 | –– |
| *Synthetic dolomite* | | | | | |
| Delle Piane et al. (2008) | (DP08) - | | | | |
| | | 873 | 300 | 4.2-5.2 | –– |
| | | 923 | 300 | 4.2-5.2 | –– |
| | | 973 | 300 | 4.2-5.2 | –– |
| | | 1023 | 300 | 4.2-5.2 | –– |
| Davis et al. (2008) | (D08) - | | | | |

[1] T = deformation temperature

[2] $P_c$ = confining pressure

[3] d/d$_p$ = grain-size/second phase grain-size

[4] $f_p$ = area/volume fraction of second phases

–– not applicable/not reported

[*] value taken from Barnhoorn et al. (2005)



| Experiment | | [1]$T$ [K] | [2]$P_c$ [MPa] | [3]d ($d_p$) [$\mu m$] | [4]$f_p$ [$\mu m$] |
|---|---|---|---|---|---|
| | | 1073 | 300 | 2.5 | $--$ |
| *Iceland spar* | | | | | |
| Wang et al. (1996) | (w96) - | | | | |
| | B3-B5 | 1053-1073 | 0.1 | 4000.00 | $--$ |
| de Bresser and Spiers (1990) | (dB90) - | | | | |
| | P2 | 1073 | 0.1 | 7000.00 | $--$ |
| *Yule marble* | | | | | |
| Heard and Raleigh (1972) | HR72| - | | | | |
| 714, 702, 708, 721, 733, 746 | | 1073 | 500 | 350.00 | $--$ |
| Heard and Raleigh (1972) | HR72$\perp$ - | | | | |
| 715, 703, 709, 722 | | 1073 | 500 | 350.00 | $--$ |
| *Carrara marble* | | | | | |
| Schmid et al. (1980) | (S80) - | | | | |
| 2736, 2663, 2695, 2868 | | 1073 | 300 | 200.00 | $--$ |
| Barnhoorn et al. (2004) | (BH04) - | | | | |
| PO344, PO298, PO266, PO264, | | 1000 | 300 | 264.00 | 0.02[*] |
| PO422, PO263, PO362, PO267, | | | | | |
| PO274, PO303, PO265, PO222, | | | | | |
| PO352 | | | | | |
| Rutter (1995) | (R95) - | | | | |
| 37, 9, 23, 28 | | 1073 | 200 | 272.00 | $--$ |
| Ter Heege et al. (2002) | (tH02) - | | | | |
| 36LM830/0.15 | | 1102 | 301 | 251.00 | $--$ |
| 36LM830/0.30 | | 1103 | 299 | | |
| 36LM830 | | 1108 | 299 | | |
| 50LM780 | | 1049 | 298 | | |
| de Bresser (2002) | (dB02) - | | | | |
| HBCM01 | | 1073 | 302 | | |

[1] T = deformation temperature

[2] $P_c$ = confining pressure

[3] d/$d_p$ = grain-size/second phase grain-size

[4] $f_p$ = area/volume fraction of second phases

$--$ not applicable/not reported

[*] value taken from Barnhoorn et al. (2005)





| Experiment | | $^1T$ [K] | $^2P_c$ [MPa] | $^3$d ($d_p$) [$\mu m$] | $^4f_p$ [$\mu m$] |
|---|---|---|---|---|---|
| | HBCM03 | 1077 | 301 | | |
| | HBCM04 | 1074 | 300 | | |
| de Bresser et al. (2005) | (dB05) - | | | | |
| | 5348 | 1073 | 300 | 150.00 | −− |
| Xu and Evans (2010) | (X10) - | | | | |
| | CM03 | 1073 | 300 | 211.12 | −− |
| Quintanilla-Terminel and Evans (2016) | (QT16) - | | | | |
| | CMhC (PI10) | 1073 | 300 | 211.12 | −− |
| *Madoc dolomite* | | | | | |
| Davis et al. (2008) | (D08) - | | | | |
| MD28E6, MD28E5, MD28E4 | | 1073 | 300 | 240.00 | −− |
| Holyoke et al. (2013) | (Hol13) - | | | | |
| | O-11 | 1073 | 300 | 240.00 | −− |
| *Synthetic calcite & alumina* | | | | | |
| Walker et al. (1990) (W90) | - | | | | |
| | C129 | 973 | 200 | 3.40 (0.25) | 0.01 |
| | C121 | 973 | 200 | 3.40 (0.25) | 0.05 |
| *Synthetic calcite & quartz* | | | | | |
| Renner et al. (2007) (R07) | - | | | | |
| | C129 | 973 | 200 | 8.20 (3.50) | 0.1 |
| *Solnhofen limestone* | | | | | |
| Schmid (1976) | (S76) - | | | | |
| 2599, 2632, 2565, | | 973 | 300 | 1-10 | −− |
| 2551, 2636, 2547, | | | | | |
| 2613, 2654, 2641, | | | | | |
| 2655, 2606, 2631, | | | | | |
| 2610 | | | | | |
| 2592, 2642, 2648, | | 1073 | 300 | 1-10 | −− |

[1] T = deformation temperature
[2] $P_c$ = confining pressure
[3] d/$d_p$ = grain-size/second phase grain-size
[4] $f_p$ = area/volume fraction of second phases
−− not applicable/not reported
* value taken from Barnhoorn et al. (2005)



| Experiment | | $^1T$ [K] | $^2P_c$ [MPa] | $^3$d ($d_p$) [$\mu m$] | $^4f_p$ [$\mu m$] |
|---|---|---|---|---|---|
| Schmid et al. (1977) (S77) - | 2559, 2561, 2662, 2646, 2549, 2557, 2608, 2668, 2554, 2591 2793, 2802, 2795, 2678, 2668, 2851, 2882, 2702, 2762 | 1073 | 300 | 4 | $--$ |
| *Lochseiten mylonite* | | | | | |
| Schmid (1982) (S82) - | | | | | |
| | | 873 | 300 | 6.5 | $--$ |
| | | 973 | 300 | 6.5 | $--$ |

$^1$ T = deformation temperature

$^2$ $P_c$ = confining pressure

$^3$ d/$d_p$ = grain-size/second phase grain-size

$^4$ $f_p$ = area/volume fraction of second phases

$^{--}$ not applicable/not reported

$^*$ value taken from Barnhoorn et al. (2005)

## 3.1 Converting reported Mg content into XMgCO$_3$

To compare data to one another we have chosen to convert all reported magnesium values to molar fractions of magnesium carbonate (XMgCO$_3$) in the following way. Firstly, if the carbonate data fall somewhere between pure calcite and dolomite, then the molecular weight of this Mg-carbonate ($MW_{Mgcc}$) can be defined as:

$$MW_{Mgcc} = ((1 - XMgCO_3) * MW_{CaCO_3}) + (XMgCO_3 * MW_{MgCO_3}) \tag{1}$$

If considering data reported as the mass fraction of magnesium ($\omega_{Mg}$), it follows that:

$$75 \quad \omega_{Mg} = \frac{MW_{Mg}}{MW_{Mgcc}} * XMgCO_3 \tag{2}$$

Substituting in equation 1 gives:

$$\omega_{Mg} = \frac{MW_{Mg} * XMgCO_3}{((1 - XMgCO_3) * MW_{CaCO_3}) + (XMgCO_3 * MW_{MgCO_3})} \tag{3}$$





Which can be rearranged to yield XMgCO$_3$:

$$XMgCO_3 = \frac{MW_{CaCO_3}}{MW_{CaCO_3} - MW_{MgCO_3} + \left(\frac{MW_{Mg}}{\omega_{Mg}}\right)} \tag{4}$$

### 3.2 Modelling XMgCO$_3$ from temperature data

The calcite-dolomite solvus as defined by Anovitz and Essene (1987) was used to convert between XMgCO$_3$ and temperature
in datasets from natural carbonates. The relationship is only valid between 473 and 1173 K.

$\quad 0 = T - A(\text{XMgCO}_3) - B/(\text{XMgCO}_3)^2 - C(\text{XMgCO}_3)^2 - D(\text{XMgCO}_3)^{0.5} - E;$

$$A = -2360.0, \quad B = -0.01345, \quad C = 2620.0, \quad D = 2608.0, \quad E = 334.0$$

### 3.3 Converting mechanical data

When data from torsion experiments are compared to other data, the reported shear stress ($\tau$) and shear strain rate ($\dot{\gamma}$) values
are converted to equivalent stress ($\sigma_{eff}$) and equivalent strain rate ($\dot{\varepsilon}_{eff}$) following the relationships outlined in Paterson and
Olgaard (2000).

$$\dot{\varepsilon}_{eff} = \frac{1}{\sqrt{3}}\dot{\gamma} \quad \text{and} \quad \sigma_{eff} = \sqrt{3}\tau$$

### 3.4 Normalisation of data

A normalisation exercise was used as the primary method of analysis. It was used to establish whether there are independent
effects of grain size and magnesium on the strain rate in carbonates. Such an analysis can obtain parameters that account sepa-
rately for any sensitivities in strain rate that exist.

For this exercise, two data sets with known grain size and magnesium contents were used: synthetic Mg-calcite experiments
from Herwegh et al. (2003) and a suite of natural samples (Wang et al., 1996; de Bresser and Spiers, 1990; Ter Heege et al.,
2002; de Bresser et al., 2005; Rutter, 1995). For the normalisation exercise we focused on data from 1073 K because this
temperature had the greatest range in grain size and magnesium across the literature.





To check the veracity of the exercise outlined above, we interrogated the results against both the data it was fit from and a wide range of other experimentally deformed carbonate with various temperatures, grain sizes, magnesium and second phase contents.

## 4 Results

Treating synthetic and natural samples separately, strain rate and stress data from deformation experiments were manipulated through multiplication of grain size and XMgCO$_3$ values to various exponents (fig. 3). The grain size (min = 14.55 $\mu m$, max = 7000.00 $\mu m$) and XMgCO$_3$ (min = 0.000013, max = 0.17) values used for each experimental data point were sourced from literature and are reported in tables 1 and 2.

Starting from

$$\dot{\varepsilon} = f(\sigma)\left.\right|_T$$

The normalisation is such that fine grained synthetic sample data is transformed to:

$$\dot{\varepsilon} d^3 X_{MgCO_3}^{-1.5} = \sigma X_{MgCO_3}^{-0.6}$$

and coarse grained natural samples becomes:

$$\dot{\varepsilon} X_{MgCO_3}^{-1.5} = \sigma X_{MgCO_3}^{-0.3}$$

For synthetic data both low and high stress domains are sensitive to grain size, while stress and strain rate require multiplication by XMgCO$_3$ to different exponents to collapse data one to one curve (fig. 3a). Natural samples are not grain size sensitive and also require separate XMgCO$_3$ exponents (fig. 3b). In both datasets the strain rate XMgCO$_3$ exponent is the same (-1.5) but is different for stress (-0.6 vs. -0.3).

These normalisation parameters can by used in conjunction with an assumed strain rate function to obtain useable flow law parameters. If the total strain rate is the sum of all rate accommodating micro mechanisms:,

$$\sum_{n=i}^{j} \dot{\varepsilon}_n = \dot{\varepsilon}_i + \dot{\varepsilon}_{i+1} + \ldots + \dot{\varepsilon}_j \,,$$

and the normalised strain rate is a function of the normalised stress, we can substitute strain rate functions for the various deformation mechanism regimes that we think are active in our data sets and invert the normalisation. In our data there are two domains in which it appears that different deformation mechanism act: a low stress domain with where data have a slope of 1 in log-log space, and a high stress domain where data have a slope of 7 in log-log space (fig. 3). For simplicity we assumed these to represent diffusion creep ($\dot{\varepsilon}_{diff}$) and dislocation creep ($\dot{\varepsilon}_{dis}$) fields and with classical temperature dependant power-law flow





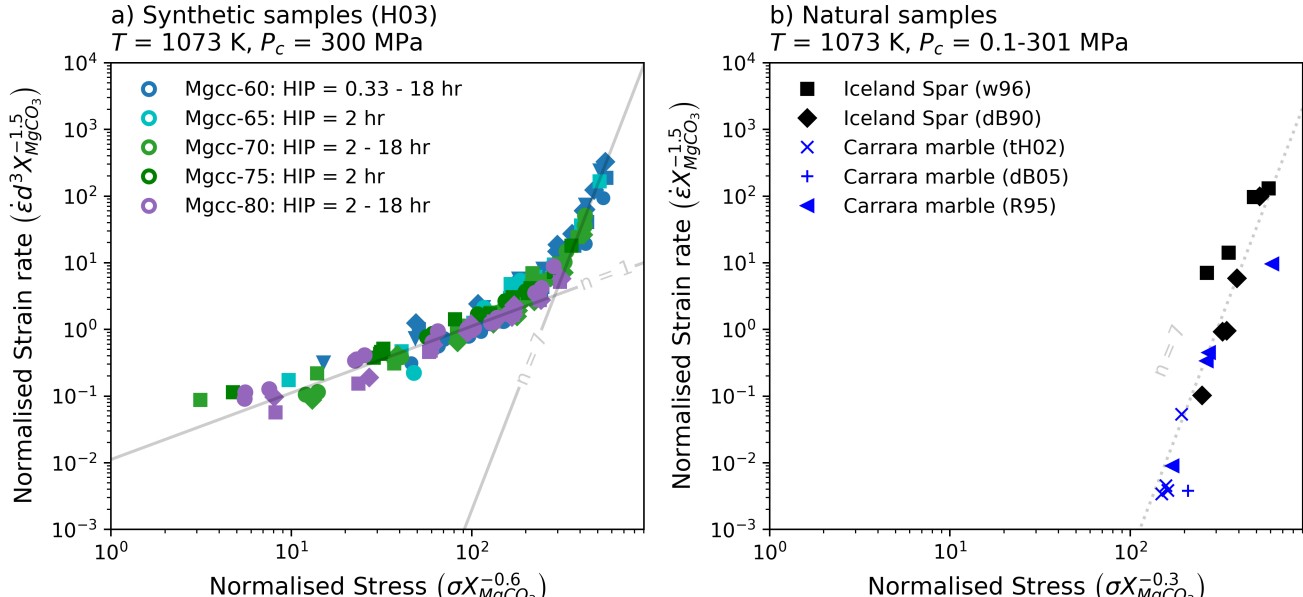

**Figure 3.** Normalisation of data.

laws.

Substituting power-law flow laws for diffusion and dislocation creep gives us the following grain size and XMgCO₃ sensi-

135 tivities:

Fine grained synthetic samples (FG):

$$\dot{\varepsilon}_{norm} = f(\sigma_{norm})$$

$$\Rightarrow \dot{\varepsilon}\, d^3\, X_{MgCO_3}^{-1.5} =$$

$$\left[ A_{diff}^{FG} (\sigma X_{MgCO_3}^{-0.6})^{n_{diff}^{FG}} \, exp\left( -\frac{Q_{diff}^{FG}}{RT} \right) \right]$$

$$+ \left[ A_{dis}^{FG} (\sigma X_{MgCO_3}^{-0.6})^{n_{dis}^{FG}} \, exp\left( -\frac{Q_{dis}^{FG}}{RT} \right) \right]$$



Coarse grained natural samples (CG):

$$\dot{\varepsilon}_{norm} = f(\sigma_{norm})$$

$$\Rightarrow \dot{\varepsilon}\, d^3\, X_{MgCO_3}^{-1.5} = \left[ A_{dis}^{CG}(\sigma X_{MgCO_3}^{-0.3})^{n_{dis}^{CG}} \, exp\left(-\frac{Q_{dis}^{CG}}{RT}\right)\right]$$

Taking the observed stress exponents of $n_{diff}$ = 1 and $n_{dis}$ = 7, then we can rearrange for the parameters:

$$\dot{\varepsilon}_{diff}^{FG} = A_{diff}^{FG} \left(\frac{d_{ref}}{d^3}\right) X_{MgCO_3}^{0.8} \left(\frac{\sigma^1}{\sigma^{ref}}\right) exp\left(-\frac{Q_{diff}^{FG}}{RT}\right) \tag{5}$$

$$\dot{\varepsilon}_{dis}^{FG} = A_{dis}^{FG} \left(\frac{d_{ref}}{d^3}\right) X_{MgCO_3}^{-2.7} \left(\frac{\sigma^7}{\sigma^{ref}}\right) exp\left(-\frac{Q_{dis}^{FG}}{RT}\right) \tag{6}$$

$$\dot{\varepsilon}_{dis}^{CG} = A_{dis}^{CG} X_{MgCO_3}^{-0.6} \left(\frac{\sigma^7}{\sigma^{ref}}\right) exp\left(-\frac{Q_{dis}^{CG}}{RT}\right) \tag{7}$$

in all cases a reference stress (1 MPa) and reference grain size (1 $\mu m$) are used to ensure that the frequency factor will have correct units for a first order rate equation (i.e. s$^{-1}$).

Three key results are established in equations 5 - 7: (1) that fine grained synthetic samples are grain size sensitive to the same exponent ($d^{-3}$) across both mechanisms; (2) that XMgCO$_3$ has opposing effects in diffusion (weakening) and dislocation creep (strengthening); and (3) the XMgCO$_3$ sensitivity of coarse grained natural samples is half as a sensitive as fine grained synthetic samples Using data from experiments to invert for the frequency factor (A) and an assumed activation energy of 200 $kJmol^{-1}$ (cf. Herwegh et al., 2003; Renner and Evans, 2002) for both diffusion and dislocation creep (see table 3), these last two points about XMgCO$_3$ sensitivity are visualised in figure 4.

Table 3: Flow law fitting parameters.

|  | A [s$^{-1}$] | n | m | u | Q [kJ mol$^{-1}$] |
|---|---|---|---|---|---|
| Fine grained (low stress) | 3.40E+07 | 1 | -3 | 0.8 | 200 |
| Fine grained (high stress) | 4.00E-08 | 7 | -3 | -2.7 | 200 |

[1] A = frequency factor      [4] u = XMgCO$_3$ exponent

[2] n = stress exponent      [5] Q = activation energy

[3] m = grain size exponent      $--$ not applicable





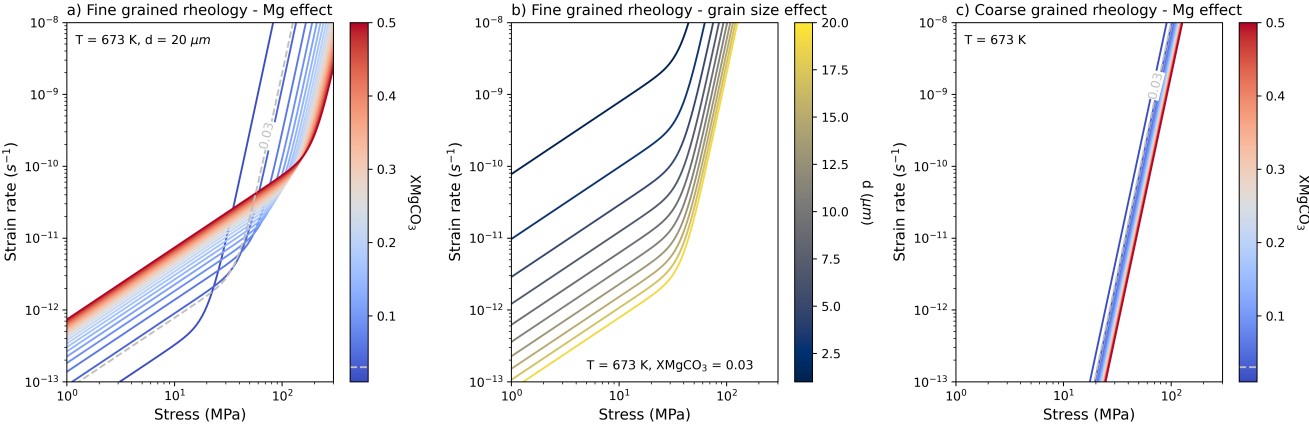

**Figure 4.** Plotting the flow laws obtained from normalisation. Curves are plotted at 673 K and the corresponding stable $XMgCO_3$ solvus value for 673 K.

|  | A [s$^{-1}$] | n | m | u | Q [kJ mol$^{-1}$] |
|---|---|---|---|---|---|
| Coarse grained | 2.16E-08 | 7 | – – | -0.6 | 200 |

[1] A = frequency factor    [4] u = $XMgCO_3$ exponent

[2] n = stress exponent    [5] Q = activation energy

[3] m = grain size exponent    – – not applicable

# 5 Discussion

The normalisation exercise revealed that the effects of grain size and magnesium can be separated. That the effect of magnesium is different for deformation mechanism domains and different between natural and synthetic starting materials. It also showed that dislocation creep in fine grained synthetic samples is grain size sensitive but that coarse grained natural samples are grain size insensitive. Most interesting of all is that magnesium switches from having a weakening effect at low stress to a strengthening effect at high stress.

The following will discuss these points and evaluate how equations 5 - 7 perform across a wide range of carbonate deformation experiments and what these new rheological laws mean for the strength of natural shear zones. In light of the apparent



divide identified in the results between fine and coarse grained carbonates, we choose to evaluate the derived equations with

data from experiments that are separated into these categories of fine and coarse grained in sections 5.2 and 5.3.

## 5.1 XMgCO$_3$ sensitivity

### 5.1.1 XMgCO$_3$ weakens a carbonate in the diffusion creep field

Firstly, the effect of magnesium on diffusion creep was noted in the original work of Herwegh et al. (2003). However, they

interpreted that magnesium indirectly affected strain rate through its control on grain growth kinetics, and by extension, grain

size. Hence it was concluded that the constitutive relation for Mg-carbonate or calcite did not require an XMgCO$_3$ term. Con-

trary to this, our new analysis suggests that the effect of magnesium can be empirically separated from grain-size. Therefore,

while the two variables are inter-dependent, we claim that magnesium has a quantifiable and direct affect on the diffusion creep

rheology of carbonates: an increase in XMgCO$_3$ weakens a carbonate experiencing diffusion creep by a power of $X_{MgCO_3}^{0.8}$

(fig. 4a).

Considering that Mg has a smaller ionic radius than Ca, it may be that the energy required to mobilise a Mg cation is lower

and hence a carbonate with more Mg may deform at a faster rate for the equivalent work done. However, this is at odds with

the observation of a broadly constant activation energies across Mg-carbonates (Herwegh et al., 2003). Suggesting that any

mechanisms being activated are thermodynamically equivalent, or possibly only one mechanism is active for all chemistries.

An alternative posit is that chemical diffusion of magnesium plays a larger role in driving cation exchange (Fisler and Cygan,

1999; Huang et al., 2008). In this case the local spatial gradients in chemistry would encourage exchange and inter-diffusion

of species. This is expected to be faster than self diffusion but has a similar activation energy (231.7 kJ/mol (Huang et al.,

2008)) as those reported for calcite cation species self diffusion or diffusion creep (Herwegh et al., 2003; Renner and Evans,

2002). If this is the case, then one should expect that as XMgCO$_3$ increases the local spatial gradients lessen and magnesium

content should have less of an effect on the strain rate. This is in fact what we observe. Figure 4a highlights that an increase in

XMgCO$_3$ has an ever diminishing affect on diffusion creep curves, which are derived from linearly spaced XMgCO$_3$ values.

### 5.1.2 XMgCO$_3$ strengthening in the dislocation field

Magnesium was previously shown to strengthen Mg-carbonates in the dislocation field (Xu et al., 2008). Specifically it was

195 demonstrated that, for a fixed strain rate, the strength of a Mg-carbonate could be described as a function of the molar percent of

magnesium to the 1/3 power. Our results agree with this and build on it (fig. 4a and c). In particular our results suggest that this

observation extends beyond synthetic Mg-carbonates to natural calcite samples. As discussed in the work of Xu et al. (2008),

the strengthening may be due to solute-drag in the glide plane or inhomogeneity in the cation structures producing obstacles

to the progress of linear defects. The second speculation may be unlikely if chemical diffusion is active, as discussed earlier

for diffusion creep, because any heterogeneity in the magnesium cation distribution would become dispersed and remove these

potential obstacles (Fisler and Cygan, 1999). In either case, increasing the molar fraction of magnesium would have the effect





hardening the carbonate. This points to a continuum in strength in dislocation creep across much of the calcite-dolomite series but the exact sensitivity seems to change between fine and coarse grained samples at high stress.

## 5.2 The rheology of fine grained carbonates

### 5.2.1 Comparing to the data it was derived from

Before comparing equations 5 - 7 to other data it is worth checking how well the equations fit the data they were derived from. Together figure 5 a and b show that a composite of the fine grained flow laws (eq.5 and 6) fit the data well and notably they perform much better than the published diffusion creep flow law of Herwegh et al. (2003). It also appears that the composite fine grained flow law performs adequately when compared to data acquired by Herwegh et al. (2003) at a lower temperature (fig. 5c). As a last comparison, a composite of the fine grained diffusion creep and coarse grained dislocation creep was compared to data collected at 1073 K (fig. 5d) to evaluate if the grain size sensitivity of dislocation creep identified in the normalisation could be modelled with the transition away from grain size sensitivity. It can be seen that a grain size intensive dislocation creep law fails to capture the behaviour of the experiments at higher stress. In total figure 5 provides confidence that equations 5 - 7 do fit the data they were derived from, being robust enough to also fit data from the same series but acquired at a lower temperature.

### 5.2.2 Comparing to other experiential data

**End-member performance: Pure Cc and Dolomite**

As a test of the limits of the normalisation results we can compare equations 5 - 6 to pure synthetic calcite (Renner et al., 2002), very low $XMgCO_3$ synthetic Mg-calcite (Xu et al., 2008) and synthetic dolomite experiments (Delle Piane et al., 2008; Davis et al., 2008). A $XMgCO_3$ minimum of 0.001 is taken as a sensible limit because data used in the normalisation is restricted to this order of magnitude.

Figure 6a shows that data from pure and very low $XMgCO_3$ experiments can be fit. Considering the uncertainties associated with grain size measurements the data are fit well. For dolomite (figs. 6b and c) lower temperature data are fit well but the relationship breaks down at higher temperature. However in the case of data from Delle Piane et al. (2008) equations 5 - 6 perform better than the flow law defined by Delle Piane et al. (2008). In contrast, the flow law of Davis et al. (2008) performs much better than our new fit.

**For different grain size controls at a fixed composition**

We identified three experiments with similar $XMgCO_3$ values that can be used to understand how well equations 5 - 6 model the rheology of carbonates with contrasting grain size controls: statically grown grain size (fig. 7a) (Xu et al., 2008), a dynamically recrystallised grain size (fig. 7b) (Barnhoorn et al., 2004) and a second phase controlled grain size (fig. 7c) (Schmid, 1982).





**Figure 5.** Plotting flow laws obtained from normalisation against data used in fit (fig. 5a); to the fit of Herwegh et al. (2003) (fig. 5b); to a lower temperature data set from Herwegh et al. (2003) not used in normalisation (fig. 5c); and a pragmatic check to see if using the obtained coarse grained flow law can explain the fine grained high stress data (fig. 5d).





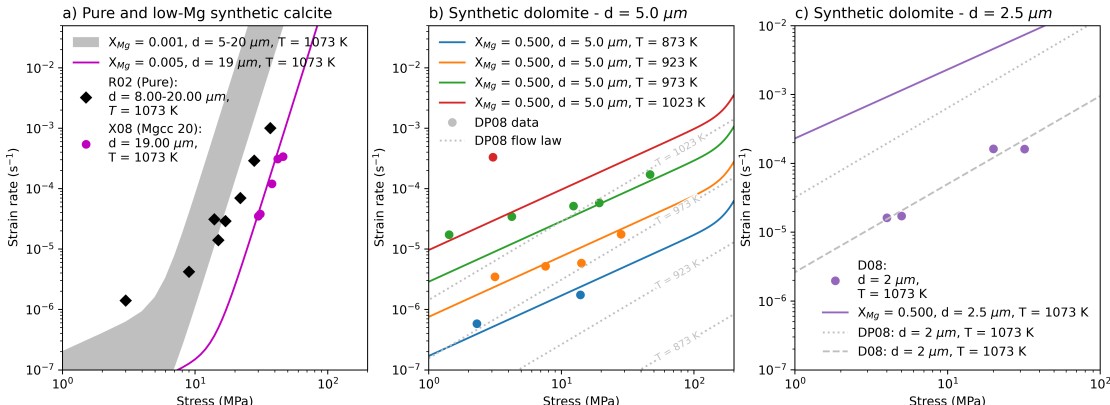

**Figure 6.** Exploring how well the fine grained flow laws obtained fit calcite and dolomite end member data.

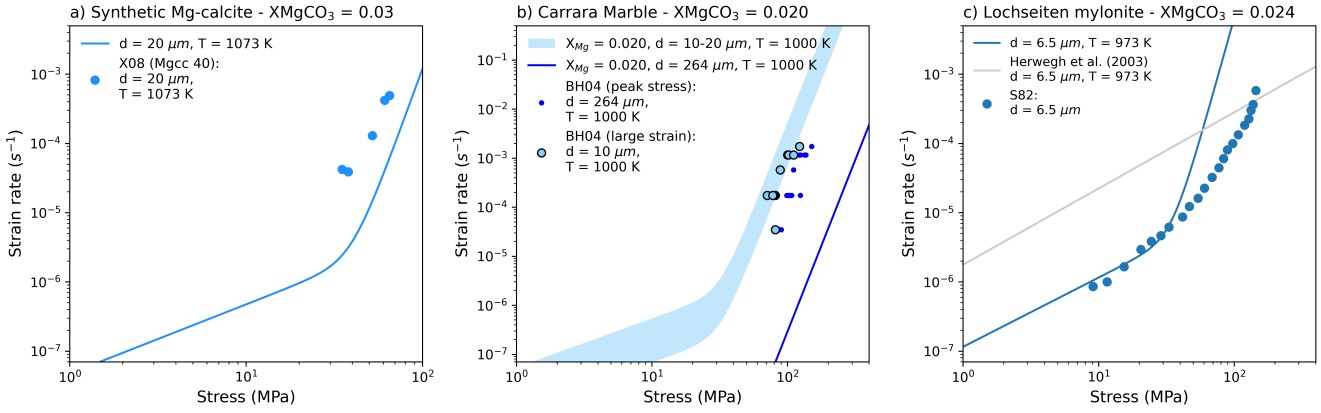

**Figure 7.** Exploring how well the fine grained flow laws fit data from experiments with stable microstructures generated with different processes that share the same magnesium content: static annealed in a synthetic sample (fig. 7a), before and after dynamically recrystallisation of a natural marble (fig. 7b), and a natural second-phase rich ultramylonite from a large shear zone (fig. 7c).

Figure 7a shows that, in line with results above, equations 5 - 6 fit synthetic Mg-cc data well. Equations 5 - 6 also perform well when compared to an experiment where the grain size evolved to stable grain size during deformation (fig. 7b). Reinforcing the framing of our discussion, the rheology of the coarser grained starting material cannot be modelled with equations 5 - 6. Lastly, it appears that the low stress rheology of a second phase controlled mylonite from the Glarus thrust can be modelled with the new laws, but the high stress behaviour deviates, possibly resulting from a very high second phase content ($f_p \leq 0.3$). Notably the newly defined XMgCO$_3$ sensitive diffusion creep law performs much better than the diffusion creep law of Herwegh et al. (2003).

**Second-phase rich carbonates**





In nature, very few carbonate mylonites consist of nominally pure calcite aggreagtes. Despite the volumetric dominance of calcite as a matrix phase, the vast majority of these mylonites contain variable amounts of other minerals, so called second phases (e.g. Ebert et al., 2007; Herwegh and Jenni, 2001; Herwegh and Kunze, 2002; Herwegh and Pfiffner, 2005; Herwegh et al., 2008). It is therefore important to evaluate the effect of such second phases on calcite rheology of mylonites, this is reflected in the partial success of equations 5 - 6 in modelling the Lochseiten mylonite data. It becomes obvious to ask how well the equations would fit carbonates with other lower second phase contents.

For this equations 5 - 6 were compared to experiments run on synthetic calcite samples with fine grained alumina added (Walker et al., 1990), Solnhofen limestone (Schmid, 1976; Schmid et al., 1977), synthetic mixtures of calcite and quartz (Renner et al., 2007) and Lochseiten mylonite data from various temperatures (Schmid, 1982). Figure 8 shows that for lower second phase contents (figs. 8a and b) data appear to be fit well. The success of equation 5 to model diffusion creep of the Lochseiten mylonite at higher temperatures is not repeated at lower temperature (fig. 8c). Figure 8d visualises how as the second phase content rises to 10% and above the rheology of transitions away from what our result can model.

## 5.3 The rheology of coarse grained carbonates

The normalisation of experimental results from coarse grained natural samples was limited to datasets where chemical and mechanical analyses were both run on the same samples. This restricted the data to two Iceland spar and three Carrara marble experiments (Wang et al., 1996; de Bresser and Spiers, 1990; Ter Heege et al., 2002; de Bresser et al., 2005; Rutter, 1995). Despite this, when compared to other coarse grained calcite experiments the data perform relatively well (9a-c) , but could not account for experiments run on dolomite (fig. 9d). In all calcite experimental cases it is clear that the range of possible mechanical fits shown with coloured regions, which are given by the available $XMgCO_3$ data (see table 1), are narrower than the the spread of data points. For some outlying data there are easy explanations. For example, in Iceland spar experiments the concentration of Mn will also influence rheology (cf. Wang et al., 1996) and for the highest stress data point of Rutter (1995) this can be explained by a marked increase in strain. However, the fit clearly overestimates the strength of Yule marble and does not capture the experimental data point of de Bresser et al. (2005) with in the range of Carrara marble possibilities. Regardless, it would appear that much of the characteristics of the rheology of coarse grained natural samples is fit well with equation 7.

## 5.4 Grain size sensitive dislocation creep?

One puzzling result is that fine grained samples are grain size sensitive in the dislocation creep field. The various underlying mechanisms documented for dislocation creep in calcite are not grain size dependent in a way that weakens the rock (cf. Renner et al., 2002). This raises several questions: is this effect real or a result of differenced in sample fabrication? Why would there be a difference between fine and coarse grained samples in dislocation creep?

Figure 10 shows data from fine and coarse grained samples that are both synthetic and natural from $XMgCO_3$ values that are close together. Data plotted from $XMgCO_3 \approx 0.07$ is made up from a fine grained synthetic starting material , Mgcc-60





**Figure 8.** Exploring how well the fine grained flow laws can fit carbonates with a variety of second-phases contents: synthetic samples doped with alumina (fig. 8a), Solnhofen limestone (fig. 8b), a natural second-phase rich ultramylonite deformed at two different temperatures (fig. 8c). Figure 8d highlights the limits of the the flow laws to model the rheology of second-phases rich carbonates by comparing them to data from a synthetic calcite-quartz mixture with no magnesium to a natural ultramylonite with magnesium.





**Figure 9.** Exploring how well the coarse grained flow law fits both the data it was fit from and other data where Mg is known for the material but not the experiment itself. Shaded areas use the max and min $XMgCO_3$ values to bracket the possible rheology.




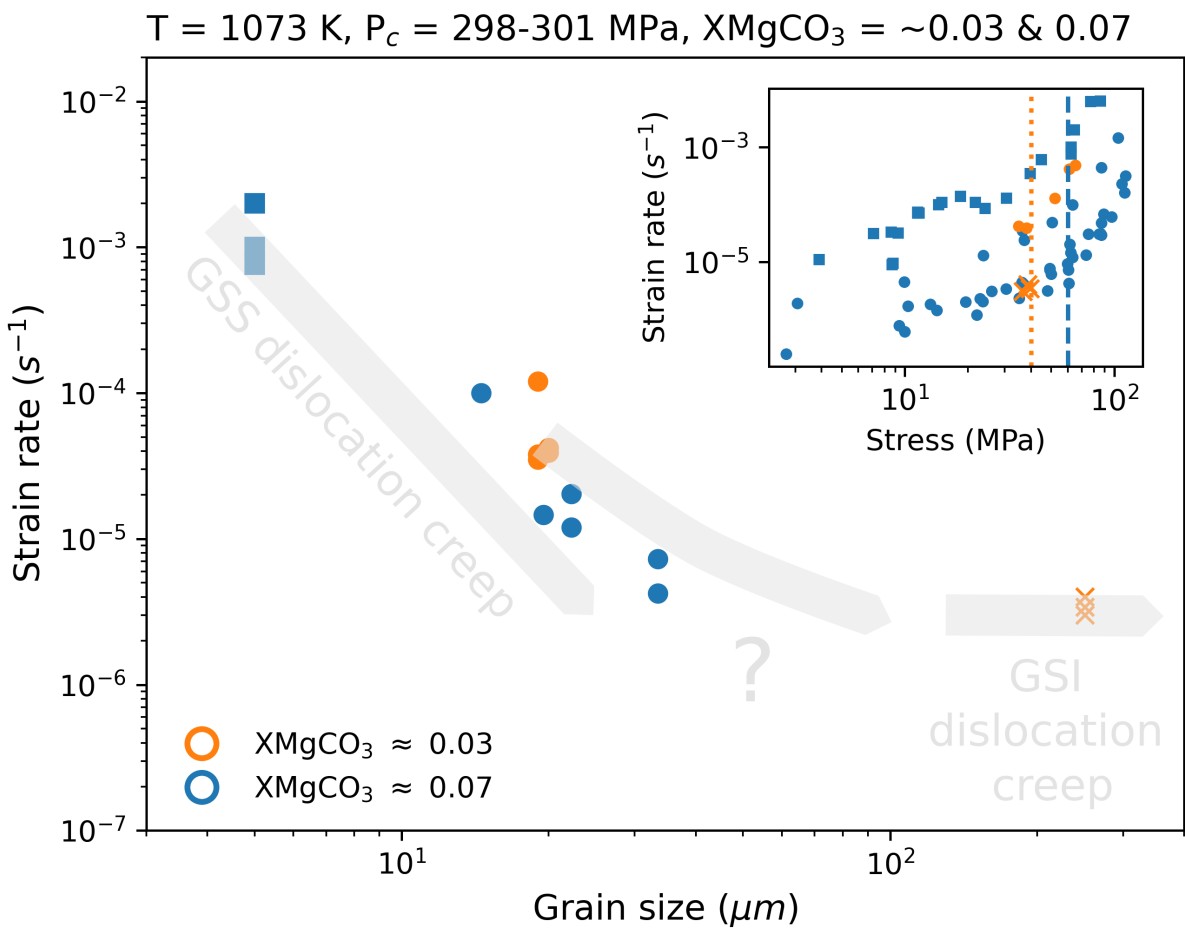

**Figure 10.** The effect of grain size on strain rate in the dislocation creep field at fixed $XMgCO_3$ values.

(Herwegh et al., 2003), and a fine grained natural starting material, Solnhofen limestone (Schmid, 1976; Schmid et al., 1977).
While data plotted from $XMgCO_3 \approx 0.03$ is made up from a fine grained synthetic starting material , Mgcc-40 (Xu et al., 2008), and a coarse grained natural starting material, Carrara marble (Ter Heege et al., 2002). Together these data highlight that: (1) the grain size sensitivity of fine grained carbonates is not restricted to only synthetic samples - it is not an artifact of Hot Isostatic Pressing (HIP) used in synthetic sample preparation; (2) there is a transition towards a grain size insensitive dislocation creep field. We conclude from this that it is a real observation that carbonates have a grain size sensitive and grain
size insensitive dislocation creep field.



Grain size sensitive dislocation creep rheologies have been phenomenologically described in other experiments run on geological materials. For example in olivine (Hansen et al., 2011, 2019), quartz (Fukuda et al., 2017), ice (Qi and Goldsby, 2021) and calcite (Renner et al., 2002; Renner and Evans, 2002; Sly et al., 2019). These grain size sensitivities have been found to
be both strengthening and weakening and have been explained in different ways. For olivine when grain size has a weakening effect ( $d^{-0.7}$, Hansen et al., 2011) a rheological model of dislocation-accommodated grain boundary sliding was proposed. In this case dislocations are thought to be generated at grain boundaries and triple junctions and as such a grain size dependence emerges. For quartz when grain size has a weakening effect ( $d^{-0.51}$, Fukuda et al., 2017) it was suggested that experiments recorded a mixed signature of end member grain size sensitive and insensitive mechanisms. In ice, grain size weakening (
$d^{-0.35}$, Qi and Goldsby, 2021) was also explained by a contribution of the dislocation-accommodated grain boundary sliding mechanism when the grain size was larger than the equilibrium subgrain size. By contrast cases where a grain size hardening effect was observed, there a modified Hall-Petch relation has been evoked: the smaller the grain size the stronger the material (e.g. Renner et al., 2002; Hansen et al., 2019). Our results do not align with any of these models, we find that the strain rate in the dislocation field is sensitive to what one would expect from a diffusion creep deformation mechanism ($d^{-3}$). Moreover,
figure 10 implies that there is a transition as grain size increases towards a grain size insensitive rheology and any underlying model would need to explain this.

For this explanation of an ever diminishing effect of grain size in the dislocation field we can turn to recent work by Breithaupt et al. (2023), who proposed a self-consistent model for deformation mediated by dislocations in which they were able
to show how a material could transition from a grain size strengthening, then weakening and ultimately grain size insensitive rheology as grain size increased. In this model the grain size sensitivity of a dislocation accommodated deformation changes because the effect of how dislocations are stored and recovered in crystals changes with grain size. Breithaupt et al. (2023) applied their model to olivine and found that it agreed with the grain size exponent measured by Hansen et al. (2011). If one extends the model of Breithaupt et al. (2023) to include diffusion creep then our phenomenological curves become more com-
parable (cf. fig 1c and fig 3.3 in Breithaupt et al., 2023; Breithaupt, 2022). Thus it might be that for the calcite experiments we revisit there is a mixed signal of diffusion creep and a grain size sensitive model for dislocations that relates not to grain boundaries as sources of dislocations, but to how dislocations are intrinsically stored and recovered in grains, and how the effects of this changes with the size of grains. The model of Breithaupt et al. (2023) also allows some explanation of the variation (hardening vs weakening) in the effect of grain size in different calcite experiments. The previous work of Renner et al. (2002)
and Sly et al. (2019) that found a strengthening effect of finer grain sizes likely capture specific conditions where diffusion creep has not been activated and hence record a domain where the Hall-Petch effect dominates (cf. fig. 7 and fig. 9b in Renner et al., 2002). For Sly et al. (2019) this is likely because experiments were run at much lower temperatures. Alternatively, for Renner et al. (2002) the difference might be due to a much finer grain size than the experiments of Herwegh et al. (2003). While the grain sizes of Renner et al. (2002) and Herwegh et al. (2003) seem cover a similar range they use different grain size
quantification methods (line intercept vs area-weighted) and if one was to convert the values of Renner et al. (2002) this might





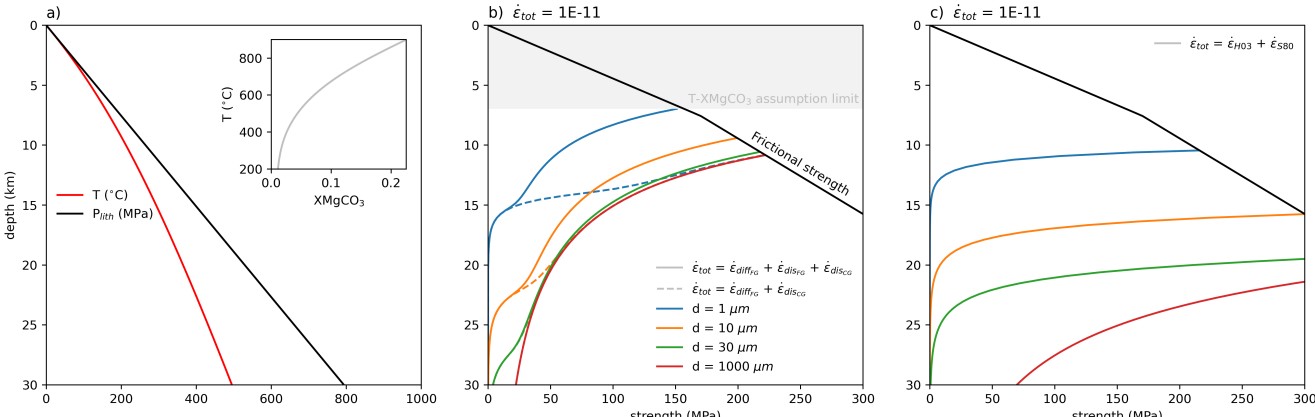

**Figure 11.** Modelling crustal strength for a given temperature-pressure-depth-$XMgCO_3$ profile (fig. 11a). Figure 11b shows the strength for various grain sizes using two combinations of the three flow laws obtained (eqs. 5 - 7). Figure 11c shows how strength changes using existing rheologies.

lead to as much as a halving of the value reported (cf. table 5 in Berger et al., 2011).

Regardless of the underlying reasons, what is clear is that our fine grained calcite data cannot be modelled well with a grain size insensitive dislocation creep rheology and some component of grain size sensitivity is needed. In the absence of adapting the model of Breithaupt et al. (2023) to include the effect of $XMgCO_3$, one could be pragmatically guided by figure 10 to explicitly model different rheologies based on grain size domains.

For example using equations 5-7,

$$\text{if d} < 30\,\mu m, \quad \dot{\varepsilon}_{tot} = \dot{\varepsilon}_{diff}^{FG} + \dot{\varepsilon}_{dis}^{FG} \tag{8}$$

$$\text{if } 30 < \text{d} < 200\,\mu m, \quad \dot{\varepsilon}_{tot} = \dot{\varepsilon}_{diff}^{FG} + \dot{\varepsilon}_{dis}^{FG} + \dot{\varepsilon}_{dis}^{CG} \tag{9}$$

$$\text{if d} > 200\,\mu m, \quad \dot{\varepsilon}_{tot} = \dot{\varepsilon}_{diff}^{FG} + \dot{\varepsilon}_{dis}^{CG} \tag{10}$$

Using an assumed geotherm, pressure with depth relation and the calcite-dolomite solvus to relate temperature to $XMgCO_3$ (equation 23 in Anovitz and Essene, 1987), figure 11 visualises this pragmatic approach (solid lines in fig. 11b) and compares it to a case where no grain size sensitive fine grained dislocation creep rheology is used (dashed lines in fig. 11b), and to the expected rheology from published flow laws for calcite (Schmid et al., 1980; Herwegh et al., 2003) (fig. 11c). The first point





of note is that, regardless of which composite rheology one chooses from our results, a XMgCO$_3$ sensitive flow law predicts much weaker carbonates with depth for a given grain size. This shows that the effect of XMgCO$_3$ must be included explicitly when modelling crustal strength. Figure 11 also highlights that if the fine grained dislocation creep rheology is accounted for, and it appears that it should as it explains much of the experimental data - including the enigmatic Solnhofen Limestone data sets (cf. 8b), then the crust becomes even weaker with depth. We know from natural shear zones that grain size also changes with temperature and so the effect shown in figure 11 will not fully capture the strength profile of carbonates.

## 5.5 Application to natural carbonate shear zones

Table 4: Naturally deformed carbonates from shear zones in the Helvetic Alps Ebert et al. (2008) and Naxos Herwegh et al. (2011)

|  | [1]T [°C] | [2]d [$\mu m$] | [3]XMgCO$_3$ |
|---|---|---|---|
| *Helvetics*[a] | 230.0 | 3.6 | 0.012198 |
|  | 239.4 | 3.03 | 0.012581 |
|  | 259.1 | 3.81 | 0.013518 |
|  | 252.0 | 5.31 | 0.013178 |
|  | 269.0 | 5.31 | 0.013987 |
|  | 271.1 | 6.49 | 0.014158 |
|  | 283.2 | 6.31 | 0.014839 |
|  | 280.4 | 8.66 | 0.014669 |
|  | 296.3 | 6.78 | 0.015670 |
|  | 318.2 | 8.41 | 0.017246 |
|  | 308.9 | 10.75 | 0.016501 |
|  | 331.8 | 10.59 | 0.018290 |
|  | 320.8 | 11.89 | 0.017395 |
|  | 316.3 | 12.06 | 0.017097 |
|  | 325.3 | 16.31 | 0.017864 |
|  | 337.8 | 15.18 | 0.018823 |
|  | 355.9 | 17.53 | 0.020506 |
|  | 358.7 | 19.39 | 0.020825 |
|  | 360.9 | 21.44 | 0.021102 |
|  | 364.6 | 22.71 | 0.021443 |

[1] T = deformation temperature [3] XMgCO$_3$ = molar fraction of
[2] d = grain size MgCO$_3$ converted from T





|  | $^1T$ [°C] | $^2$d [$\mu m$] | $^3$XMgCO$_3$ |
|---|---|---|---|
|  | 371.3 | 27.78 | 0.022210 |
|  | 337.8 | 29.85 | 0.018823 |
|  | 342.6 | 31.62 | 0.019270 |
|  | 364.6 | 34.97 | 0.021443 |
|  | 363.1 | 38.68 | 0.021336 |
|  | 380.4 | 39.81 | 0.023232 |
|  | 395.4 | 50.85 | 0.025107 |
| *Naxos$^b$* | 390.0 | 130.78 | 0.024468 |
|  | 390.0 | 155.74 | 0.024468 |
|  | 470.0 | 192.55 | 0.037206 |
|  | 500.0 | 373.09 | 0.043512 |
|  | 500.0 | 305.56 | 0.043512 |
|  | 560.0 | 1132.92 | 0.058913 |
|  | 650.0 | 2250.63 | 0.089844 |
|  | 650.0 | 1710.26 | 0.089844 |
|  | 700.0 | 2852.89 | 0.111487 |
|  | 700.0 | 2222.71 | 0.111487 |
|  | 700.0 | 2062.35 | 0.111487 |
|  | 700.0 | 2250.63 | 0.111487 |
|  | 700.0 | 1668.1 | 0.111487 |

[1] T = deformation temperature     [3] XMgCO$_3$ = molar fraction of

[2] d = grain size     MgCO$_3$ converted from T

Using temperature and grain size measurements made in mylonites from natural carbonate shear zones we can account for the combined effect of grain size and XMgCO$_3$ on strength. Figure 12 shows data from the various thrusts in the Helevtic Nappes (Ebert et al., 2008) and from Naxos (Herwegh et al., 2011) that covers greenschist to amphibolite facies conditions in carbonate shear zones. As before, the calcite-dolomite solvus can be used to relate temperature measurements to inferred XMgCO$_3$ values. For the purposes of visualisation figure 12 shows moving average curves for the various rheological models but only shows data points for one rehology - the pragmatic composite rheology, eg. eqs. 5-7.





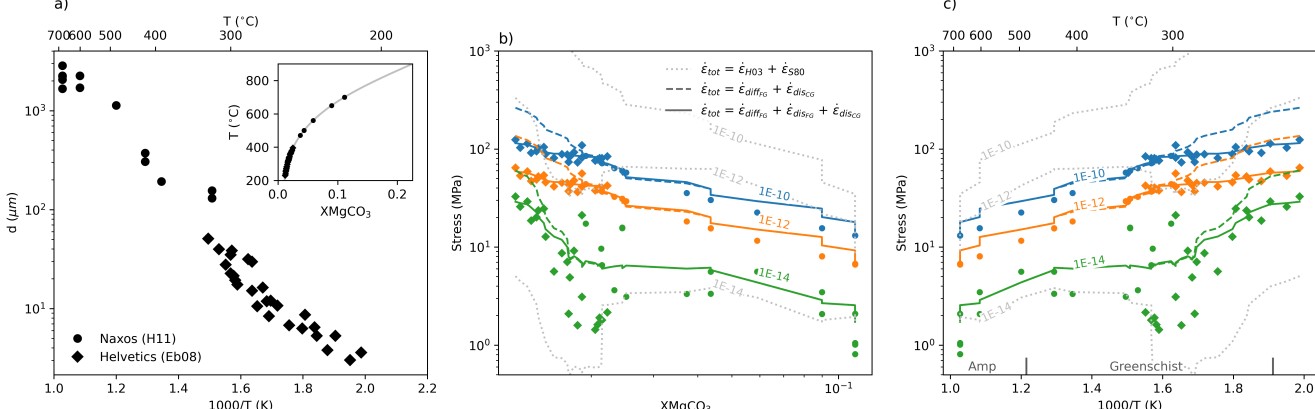

**Figure 12.** Modelling crustal strength using real grain size and temperature data (fig. 12a). $XMgCO_3$ values are derived from temperature values (inset in fig. 12a). Figure 12b and c show the stress expected for a variation in $XMgCO_3$ and temperature using two combinations of the three flow laws obtained (eqs. 5 - 7). Existing rheologies are shown for comparison as dashed grey lines.

Most notably for faster strain rates in lower greenschist facies conditions the strength of carbonates with depth flattens out. This occurs because of the competing effects grain size and magnesium at these conditions (weakening - $\propto d^{-3}$ & $X_{MgCO_3}^{-2.7}$; strengthening - $\propto X_{MgCO_3}^{0.8}$). Carbonate rheology in these shear zones seems to balance out the effects of moving from smaller to larger grain sizes and an increasing XMgCO$_3$ content. For higher temperatures it is expected that a shear zone will widen, and therefore strain rate will decrease, making it hard to compare data from the Helvetics and Naxos together. Field obser-

vations from the Helvetic nappes suggest that, when accounting for shear widths, strain rates are relatively constant and can be reasonably modelled with values of between $1\text{x}10^{-10}$ and $1\text{x}10^{-11}$ (Herwegh and Pfiffner, 2005). Following from this if we only use data from the Helvetics and assume a strain rate of $1\text{x}10^{-11}$ we can more carefully consider the strength of a crustal scale carbonate shear zone. Using the same data as previous, figure 13 presents only data from the Helvetics. The blue and orange diamond markers give upper and lower bounds on crustal strength for rheologies that are grain size and XMgCO$_3$

sensitive. If fine grained shear zones deform with rheology that includes a grain size sensitive dislocation creep flow law (blue diamonds) then the crustal strength becomes very close to constant for an interval of roughly 200 °C. This is comparable to estimates that come from accounting for grain size distributions (crosses in fig. 13). If however, the rheology is considered to be a composite of a grain size and XMgCO$_3$ sensitive diffusion creep mechanism and a XMgCO$_3$ sensitive dislocation creep mechanism (orange diamonds) then a shear zone would have a larger strength evolution with increasing temperature but it

would still be weaker than predicted by piezometric relations.





**Figure 13.** A crustal strength profile constructed with only the Helvetic thrust data set (Ebert et al., 2008). Strength estimates from a grain size paleopiezometer (Barnhoorn et al., 2004) and from naturally constrained rheological modelling that includes grain size distributions (Herwegh et al., 2005) are plotted for comparison.



## 6    Summary and outlook

We have brought together much of the existing literature of carbonate deformation experiments run at high homologous temper-
atures and found that most data can be explained well by the distinct effects of grain size and the molar fraction of magnesium
carbonate. For a given grain size, the above results highlight that there is a quantifiable continuum in strength of carbonates
with magnesium content and that the role of magnesium changes with deformation mechanism (fig. 4a). Furthermore for a
given molar fraction of magnesium, our review of the data show that at high stresses, at which dislocation mediated deforma-
tion will occur, grain size has a weakening effect (fig. 10). This is contrary to the current use of the modified Peierls law for
calcite dislocation creep rheology, where grain size has a hardening effect.

Together these new results suggest that the deformation of magnesium carbonates should be considered in a more unified
fashion. For low stress diffusion creep behaviour it might be more pragmatic to view these carbonates as existing in a contin-
uum of material strength and not as distinct materials. This is not true for the dislocation creep field where dolomite seems to
not conform to predictions of a continuous change in strength with magnesium. In cases of mixed grain/clast populations of
calcite and dolomite one should still identify both the phase proportions and $XMgCO_3$ of equilibrated dolomite and calcite if
one wishes to calculate a mixed rheology (e.g. Dimanov and Dresen, 2005; Renner et al., 2007).

The biggest limitations of our contribution is that we cannot provide an underlying microphysical model. This is particu-
larly challenging because our data show that the underlying deformation mechanisms need to account for both the effect of
magnesium and a grain size weakening dislocation creep. The best starting point would appear to be to build on the work of
Breithaupt et al. (2023) and include a physically sensible magnesium dependence.

A main take away from this work is that even small differences in magnesium can have a large effect, especially when
compounded by the role of grain size. Our discussion explored how crustal strength might significantly deviate from the
current expectations, producing a very weak and almost uniform rheology with temperature and therefore depth (fig. 13).
If true, this removes any usefulness of carbonate paleopiezometery for estimating crustal strength. More importantly, these
results could extend to the olivine system where it has also been shown that there is a change in strength with an increase
in iron concentration (Zhao et al., 2009, 2018). As such our work highlights the need to better integrate the effects of solid
solutions in important Earth materials.

*Code and data availability.* A python script for the conversion of magnesium data and a csv file of the mechanical data reviewed in this
publication are provided in the supplement.



*Author contributions.* JG and MH designed the study. JG preformed the data collation and analyses. All authors were involved in the interpretation of the results and the writing of the final manuscript.

*Competing interests.* The authors declare that they have no conflict of interests.

*Acknowledgements.* Jörg Renner is thanked deeply for his contribution to the discussion of data over the course of the many years it took to bring this publication together. This research has been supported by the Schweizerischer Nationalfonds zur Förderung der Wissenschaftlichen Forschung (grant no. 162340) and the Leverhulme Trust through the Leverhulme Early Career Fellowship.



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
