# Peer review of "On unifying carbonate rheology"

_EGUsphere, 2025_

## Referee Comment (RC3)

Review of "On unifying carbonate rheology"

Submitted to EGUSphere
by J. Gilgannon and M. Herwegh
* * *
This manuscript presents an overview of the many papers published on high temperature flow laws for carbonates, to a great extent made up of natural and synthetic polycrystalline calcite, and the compiled results are used to evaluate a global flow law that is consistent with all of the experimental data. While earlier studies attributed changes in rheology to grain size or to composition (primarily $X_{MgCO3}$) alone, this re-evaluation of the entire dataset shows that polycrystalline calcite depends on both parameters, including deformation dominated by dislocation creep and diffusion creep. That is, deformation due to intracrystalline and inter-crystalline mechanisms depend on these parameters. This result is new and significant, particularly as increasing numbers of grain-size sensitive flow laws for geologically important lithologies (carbonates and silicates) are published and assumed to be due to grain boundary diffusion and sliding, and dislocation processes are typically thought to be independent of grain size (when internal climb and recover preclude increases in defect density) or depend on grain size when glide (Peierls law deformation) occurs with limited diffusion (and climb), giving rise to Hall-Petch strengthening by grain boundaries, not the weakening determined by this analysis.

One detail that might help the reader follow the development here is to state which carbonate minerals are included in the analysis. I find the inclusion of dolomite in this paper interesting, as some would not expect a single flow law to apply to both calcite aggregates/rocks (of varying but relatively low solid solution $X_{MgCO3}$ values) and dolomite (at large $X_{MgCO3} = 0.5$). At the same time, I think this is an interesting perspective brought out by including both carbonate phases, calcite and dolomite, and attempting a fit to the entire dataset. At the outset, the different slip systems of these two carbonates would be expected to lead to different dislocation creep laws. However, without further information about the grain boundary structures and properties, I see the authors' point of including both fine-grained calcite and dolomite data that are dominated by grain-size-sensitive processes, including grain boundary diffusion and sliding. Why not compare these results normalizing by parameters of grain size and $X_{MgCO3}$? It is worth stating the beginning premise(s) of this analysis (and why you include dolomite data). Taking this further, it might be interesting to include magnesite deformation data (with $X_{MgCO3}$ nearly equal to 1.0; Holyoke et al., 2014). Indeed, the mineral symmetry and structure of magnesite is essentially the same as calcite, while dolomite differs due to its ordered cation structure. Thus, any trends that go beyond creep of an individual carbonate phase might be more likely to hold for calcite and magnesite than for calcite and dolomite.

Of course, when applications are made to crustal rocks, the deformation behavior of calcite (of varying Mg content) and dolomite will be most useful. Thus, the ability to formulate a flow law that fits such a large dataset for crustal carbonates at varied conditions, grain sizes, and compositions is impressive and will serve as a reference in applications to natural deformations of carbonates. This contribution should therefore be of great interest to readers of EGUsphere, including geologists studying deformation in the laboratory, natural shear zones, and theoreticians evaluating physical processes of deformation.

This manuscript is well written and shouldn't take much revision for final publication. In the following, I offer some minor points that may improve the manuscript. However, I leave much of this to the authors' discretion.

Early in Section 3.4 (Normalization of data), it would be worth stating more explicitly that all mechanical data sets where calcite compositions have been reported and dolomite have been used in the normalization. Or I may have misunderstood how this was done. The dolomite data may throw off a best-fit to calcite aggregate data. In any case, please state exactly how this was done.

In Figure 11, it would be worth stating what the assumed tectonic setting is. These are classic lithosphere-dimensioned yield envelopes for carbonate deformation including frictional and plastic flow laws for an assumed, fixed strain rate, geothermal gradient and tectonic stress state (compressional, extensional, strike-slip/transform boundaries). Please include the assumed tectonic setting in the caption. Similarly, please include the assumed strain rate in figures 12b and 12c.

I find the comparisons of flow laws for non-zero-$X_{MgCO3}$ calcite aggregates and dolomite very interesting, and I wonder if the authors might add some possible explanations of the differences in the discussion of this manuscript. At the outset, differences in crystal symmetry and structure of calcite and dolomite, and the different slip systems of these two minerals offer obvious reasons that their flow laws for coarse-grained aggregates associated with intracrystalline deformation cannot readily be combined into one simple relation. However, it is fair to wonder why this doesn't work well for grain-size-sensitive deformation and processes at grain boundaries (grain boundary diffusion and sliding). We know little about the atomic structures of grain boundaries, so a relationship that includes grain size and $X_{MgCO3}$ might be expected to work for inter-crystalline creep mechanisms in relatively "unstructured" carbonate grain boundaries. It could be that mean jump distances or frequencies might differ for dolomite grain boundaries (where dolomite grains are ordered in Ca and Mg) from those at calcite grain boundaries. This might mean that grain boundary diffusion depends more strongly on $X_{MgCO3}$ than captured by grain-size-sensitive calcite deformation results. Alternatively, nucleation (or reaction to add to one of the dolomite grains adjacent to a grain boundary) during diffusion creep and grain boundary sliding may differ at $X_{MgCO3}$ values near 0.5. I don't think these potential differences can be evaluated without further data or information, but they are interesting possibilities. I wonder if grain-size-sensitive deformation of magnesite (at

$X_{MgCO3} =1$) with the same mineral structure as calcite might fit the universal carbonate deformation law presented here better than flow laws reported for dolomite.

Throughout, this manuscript is well written and I have only very minor editorial suggestions for rewording, which I will list below by line number:

Line 2 (abstract) – "… limestones and marbles.  Such an"

Line 100 – Was this normalization done just for calcite samples?  Or both calcite and dolomite samples?  Please specify explicitly ("… of natural calcite samples… " or "… of natural calcite and synthetic (?) dolomite  samples …")

Line 108 – again please specify if just calcite samples or both calcite and dolomite.  The max $X_{MgCO3}$ value of 0.17 certainly suggests that dolomite data were not included in the global fit.

Line 115 – "… sample data are transformed to:"

Line 117 – "and results for coarse grained …"

Line 120 – "… exponents to collapse data to a one-to-one curve …"

Line 128 – "… regimes that we think contribute to deformation and can be used in investigation and normalization.  In our data there are two"

Line 129 – "… different deformation mechanisms are apparent:  a low stress domain in which data have a slope of 1 in"

Line 130 – "… a high stress domain characterized by a slope of 7 …"

Line 147 – delete "then"

Line 155 – suggest deleting "that"

Line 156 – "… mechanisms; (2) XMgCO3 has two opposing effects …"

Line 157 – "… coarse grained natural samples is only half that of the compositional sensitivity of fine grained"

Line 160 – suggest replacing "visualized" by "illustrated"

Line 171 – "… and coarse grained samples in sections 5.2 and 5.3."

Line 174 – "The effect of magnesium on diffusion creep was first noted in the original work …"

Line 177 – "… magnesium can be separated empirically from grain size. Therefore,"

Lines 184-185 – "the observation of the same or broadly similar activation energies across Mg-carbonates (Herwegh et al., 2003). This observation suggests that mechanisms activated are thermodynamically equivalent, possibly the exact same mechanism for all chemistries."

Line 189 – "2008)) as reported for cation species self diffusion or diffusion creep of calcite (Herwegh …"

Line 190 – "… one should expect that local spatial gradient are reduced as XMgCO3 increases, and magnesium"

Line 191 – "… This is what we observe. …"

Line 192 – delete "which are derived from linearly spaced XMgCO3 values"

Lines 196-197 – "… (fig. 4a and c). Indeed, our results suggest that this compositional effect on flow law extended beyond synthetic Mg-carbonates …"

Line 198 – "strengthening by XMgCO3 may be due to solute-drag …"

Line 199 – "to the progress of linear defects. This second possibility is not expected if chemical diffusion is substantial as discussed earlier"

Line 235 – delete "the framing of"

Line 238 – "0.3). Notably the newly defined XMgCO3-sensitive diffusion creep law …"

Lines 245-246 – "… Lochseiten mylonite data. It is important to ask how well the equations fit carbonates …"

Line 248 – "Equations 5-6 were compared …"

Lines 251-253 – "… data appear to be fit well. In the case of diffusion creep of the Lochseiten mylonite, the data fit equation 5 at higher temperatures but not at lower temperatures (fig. 8c). Figure 8d shows that, as the second phase content rises to 10% and above, the observed rheologies depart from our model."

Line 255 – "The normalization of experimental results of coarse-grained natural sample results was limited to …"

Line 258 – "When compared to other coarse grained calcite experiments, the data fit the model relatively well (9a-c). However, they do"

Line 261 – "the spread of data points. …."

Line 262 – delete "will"

Line 263 – "there is a marked increase in strain. 'the fit clearly overestimates …"

Line 264 – "… with in the range of Carrara marble experimental results. Regardless"

Line 269 – "… result of differences in sample …"

Line 271 – "… synthetic and natural for XMgCO3 values that"

Line 272 – "… Data plotted for SMgCO3 = 0.07 come from a fine grained …"

Line 275 – "While data plotted for XMgCO3 = 0.03 comes from a fine grained …

Line 277 – "… is not restricted to synthetic samples alone – it is not an artifact"

Lines 278-280 – "… there is a transition to a grain size sensitive dislocation creep field. We conclude from this that grain size sensitive and grain size insensitive dislocation creep fields are real."

Line 292 – "a modified Hall-Petch relation has been proposed; the smaller the grain size, the stronger …"

Lines 294-296 – "in the dislocation field appears to be similar to the $d^{-3}$ dependence of diffusion creep. Moreover, figure 10 implies that there is a transition as grain size increases to a grain size insensitive rheology."

Line 298 – "To explain the ever diminishing …"

Line 300 – "… could transition from grain size strengthening to weakening and …"

Lines 301-302 – "deformation as grain size increased. In this model the grain size sensitivity of dislocation accommodated deformation changes according to the storage and recovery of dislocations, and the dependence of these processes on grain size. Breithaupt et al. (2023)"

Lines 305-309 – "… Breithaupt, 2022). Thus, the apparent results for diffusion creep and grain size sensitive deformation of polycrystalline calcite might be linked to the storage and recovery in grains, and how these change with size of grains. The model of Breithaupt et al. (2023) also explains the variation (hardening and weakening) in the effect of grain size for different calcite experiments. …"

Lines 312- 313 – "et al., 2002). This is most likely for lower temperature experiments of Sly et al. (2019). While the grain sizes of Renner et al. (2002) and Herwegh et al. (2003) seem to cover a similar range …"

Lines 315-316 – "… vs area-weighted); grain size values of Renner et al. (2002) might be half those measured by Herwegh et al. (2003; cf. table 5 in Berhger et al., 2011)."

Line 319 – "… sensitivity is needed.  Short of adapting"

Lines 320-321 – "… the effect of XMgCO3, a pragmatic guide to modeling different rheologies based on grain size domains (Fig. 10)."

Line 323 – replace "if" by "for"

Line 325 – replace "if" by "for"

Line 326 – replace "if" by "for"

Line 328 – "… 1987), figure 11 illustrates this approach …"

Lines 329-331 – "it to a case for fine grained dislocation creep rheology that shows no grain size effect (dashed lines in fig. 11b), and the published flow laws for calcite (Schmid et al. 1980; Herwegh et al., 2003) fig. 11c). Regardless of which composite …"

Lines 333-336 – "when modelling crustal strength.  Figure 11 also shows that the fine grained dislocation creep rheology predicts low strengths in the crust with depth."

Line 338 – "Using temperature and grain size measurements for mylonites …"

Line 339 – "… Figure 12 shows data from several thrusts of the Helvetic"

Lines 340-341 – "… Naxos (Herwegh et al., 2011) deformed at greenschist to amphibolite facies conditions.  As before, the calcite-dolomite solvus …"

Line 353 – "… Using the same data as previously, figure …"

Line 355 – "… shear zones deform by a grain size sensitive …"

Line 359 – "… a shear zone would show more strength evolution …"

Line 367 – I'm a bit confused by the use of "grain size has a weakening effect". This would seem to state that larger grain sizes weaken the flow law, but I think you mean to state that fine-grain sizes (or larger grain boundary densities) weaken the flow law. Please revise.  How about "grain boundaries have a weakening effect"?

Line 382 – "An. Important conclusion of this work is that …"

Line 385 – "If true, this calls into question the predictions of carbonate paleopiezometry for crustal strength.  More importantly, these"

Lines 386-387 – "… also been shown that flow strength depends on iron concentration (Zhao et al., 2009-2018).  …"

I look forward to seeing this paper in print,

Andreas Kronenberg

---

## Author Comment (AC1)

Dear Editor,

We kindly thank the three reviewers for their comments on our manuscript.

One common thread in their different discussions of the text is that we were a bit too quick in the step regarding our method and the description of figures. The primary changes in the manuscript focused on addressing these aspects and the issues with clarity that they brought.

Beyond this, Referee 2 noted some additional limitations to the interpretations we make and we now include those in the discussion. Additionally, Referee 3 drew our attention to a comparison to magnesite (the Mg end member of the carbonate system we investigate) and we have now also incorporated data from relevant experiments and discuss them. In both case, we thank the reviewers for drawing attention to literature we missed.

Lastly, during the review of the initial submission, Mario Ebel noted that figure 11, now figure 12, was problematic for readers with colour vision deficiencies. We have changed this figure to now label the lines with the model grain sizes. This should now cover all of the colour vision deficiencies, including Monochromacy/Achromatopsia.

We hope that the changes to the manuscript increase its clarity and scope.

Best,

James Gilgannon

In the follow rebuttal we have addressed each reviewer in turn. First we address the general points they raised and then the specific comments below that. We only included the relevant sections of the referee's comments and have coloured them in a light grey to make it easier to read our replies.

**Referee 1**

General comments

Specifically, I would like to learn more about the central methodological part of the manuscript - the normalization of the data (Sections 3.4 and 4). The procedure of normalization is the key part of the study, but its description is very short. Where do the values of exponents used in the normalization come from? Did you test different values? How did you choose these particular values? What is the estimated error range?

You are right the text is too short. We have changed the text accordingly. For the purpose of replying to your questions we have also briefly responded here as well.

The choice of exponents was as follows:

1. Grain size sensitivity was fixed at 3 in line with the microphysical model for grain boundary diffusion.
2. XMgCO3 sensitivity was varied freely on both axes systematically. There was no statistical or mathematical basis for this iteration and convergence: it was user defined.
3. Stress sensitivity was a result of the normalisation in strain rate vs. stress space.

We did test different values as part of the user guided iteration toward data convergence on one 'master line'. A more sophisticated method would have been to automatically iterate through exponent values and find the best fit via a regression but we did not do this.

We do not have any estimated errors for the parameters, mostly because we cannot calculate through anything meaningful from the various data sources. If one was to run new experiments, then one might try and use uncertainties in the chemical measurements. One might also try and understand if the magnesium sensitivity is bundled, as we assume in our work, out of the frequency factor or if it is the mechanism represented in the activation energy that is sensitive. This would require propagation of error across the various sensors on the apparatus and the measurements of chemistry. Again, if we had used a more sophisticated exponent iteration that looked for a minimum in residuals then we could maybe give an empirical fitting uncertainty, but as we did not do this, the values must stand with out an error.

Specific comments

I also often miss information in figure captions. Namely:
Fig. 1 - what is the meaning of different colors? Is the same color coding also used in other figures in the manuscript? Abbreviations (H11, Eb08…) are not explained in the caption - a reference to Tables would help.
We have changed the caption accordingly.

Fig. 2c - description of this panel is not clear. What is in the inset panel? How is it related to the panel c)?
We have changed the caption accordingly.

Fig. 3! - The figure is busy and important at the same time, but the caption is practically lacking.
You are right, the caption only gives a very curt description. We have changed it to better explain what the reader is seeing.

Fig. 4 - What are the individual lines/curves? What is their spacing?
We have changed the colour bar to show ticks for the discrete values of the curves for clarity.

Fig. 10 - What is the relationship between the inset panel and the main panel? The caption is, again, too brief.
We have changed the caption to better guide a reader but we do point out that the relationship between the three dimensions of strain rate, stress and grain size are made explicitly clear through the constitutive relationship of the rheological relationship.

Fig. 11a - Can you plot the depth-XMgCO3(T) relationship directly in the main panel (having a secondary x-axis)?

We do not think that this add to the figure. We prefer to retain the convention of the solvus plot shown in the inset.

Fig. 11b,c - References to the flow laws are missing in the caption.
We have changed the caption accordingly.

Fig. 12 - Which existing rheologies? Line 342 can be moved into the caption.
We have changed the caption to better explain what we refer to.

Minor comments:
Within captions, panels can be referred to as a) or b) and not Figure 1a or Figure 1b.
We choose to retain our formulation.

Some grey lines in the figures are barely visible.
We have looked through the figures and find them to be legible. With out more specific direction we retain their grey colour as we feel that it keeps the various figures' emphasis balanced.

Equations in sections 3.2, 3.3 -  colons before the equations are missing?
We feel the that the formatting is correct and will take direction of the editing team as we go further towards publication.

line 217 - Why do you write pure "Cc" here? Is it explained anywhere?
The text has been changed accordingly.

line 228 - This heading is not clear to me
We have changed the text to:
"Rheology of samples with different grain size controlling mechanisms"

Lines 345-355 are difficult to follow
We have changed the text to:
"Most notably, for faster strain rates (1E-10 and 1E-12) at lower greenschist facies conditions the strength of carbonates flattens out with depth."

There are a lot of typos and small grammatical mistakes - a more careful reading and writing is needed.
The text has been changed accordingly.

**Referee 2 - Brian Evans**
General comments

In calcite rocks (and metals), twinning causes a similar effect by dividing the grains into separate slip domains. This inverse relation between twinning dimension and strength (the TWIP effect) is well-established (although the details are still a topic of discussion) [Rowe and Rutter, 1990; Rutter et al., 2022; Rybacki et al., 2021]. In the case of twinning, there is no change in chemistry during the deformation, and so, the strength effects are due solely to structural changes in the grains. Thus, I would be reluctant to say that reduction in grain (twin) size has a weakening effect on calcite rocks under all conditions.

This is a fair point and we have noted in the summary this limitation.

Similarly, in the discussion, you have extended the findings to conditions of T and rate in the Earth that are quite different from the experimental conditions (line 385 and following). This seems risky, particularly given that at least some of the data are derived from the transition between grain boundary sliding/ diffusion creep and dislocation creep. If the dependence on grain size changes with changes in T and rate as it does in metals, then it is more likely that there are two regions with different behavior, much the same as is proposed for deformation maps.

We have formulated this more carefully in the text now.

Specific comments

In abstract: Sentence beginning with "Most notably our results…can be shown to have a weakening effect in dislocation creep…" change to "in high temperature creep where diffusion is important…" Change sentence with "This is the opposite finding to the currently accepted flow law" to acknowledge the fact that we might expect there to be two regimes with different dependences on grain size.

We have changed the text to try and reflect the referee's comments. While we agree, and discuss where the hardening effect may exist in the paper, we think that for an abstract we wish simply to state the fact that our results draw an opposite conclusion to previous work which analysed similar temperatures and rates.

Figure 1 and Table 2: The data sets are identified by codes (H03) and so on. But the codes are hidden in table 2; make a new column or include them in the first column so that they are more prominent. In the figure captions (Particularly figure 1) provide a reference to table 2 so that the reader will know what the codes mean.

We have changed the table 2 to make the abbreviations for experiments more prominent and added references in figure captions.

The description of the normalization techniques wasn't at all clear. Was the success in normalization judged solely by observing a decreased dispersion of the data points? Did you quantify the normalization by some statistical measure? I am assume that the normalization technique robustly eliminates any cross-correlation between the grain size and the magnesium content.

This is something that Referee 1 also picked up on. We have added much more text and an example figure to more explicitly state what was done. Please refer to our answer to Referee 1.

Is it correct that there are 6 components (Afg, nfg, Qfg, Acg, ncg, Qcg)) to be determined for fine-grained samples, and three more to be determined for the coarse-grained samples? Given this number of variables can you be assured that you will achieve accuracy? I apologize if this question is ill-informed, but perhaps a comment in the text is in order.

There is nothing to apologise for, it is not an ill-informed question but a question that probes at one of our biggest limitations. We have added to the text to clarify this limitation more. Also please see our answer to Referee 1 on this matter as they raised it too.

**Referee 3 - Andreas Kronenberg**

General comments

One detail that might help the reader follow the development here is to state which carbonate minerals are included in the analysis.

The text has been changed accordingly and direction to the details of this change can be found in our replies to the referee's specific comments below.

Why not compare these results normalizing by parameters of grain size and XMgCO3? It is worth stating the beginning premise(s) of this analysis (and why you include dolomite data). Taking this further, it might be interesting to include magnesite deformation data (with XMgCO3 nearly equal to 1.0; Holyoke et al., 2014). Indeed, the mineral symmetry and structure of magnesite is essentially the same as calcite, while dolomite differs due to its ordered cation structure. Thus, any trends that go beyond creep of an individual carbonate phase might be more likely to hold for calcite and magnesite than for calcite and dolomite.

We appreciate the referee pointing us to the work by Holyoke et al. (2014) on magnesite as we had some how missed it in our review the literature. We have added the magnesite data and expanded on the discussion where relevant. The details of where this was added are noted below in the responses to specific comments.

Specific comments

This manuscript is well written and shouldn't take much revision for final publication. In the following, I offer some minor points that may improve the manuscript. However, I leave much of this to the authors' discretion.

Early in Section 3.4 (Normalization of data), it would be worth stating more explicitly that all mechanical data sets where calcite compositions have been reported and dolomite have been used in the normalization. Or I may have misunderstood how this was done. The dolomite data may throw off a best-fit to calcite aggregate data. In any case, please state exactly how this was done.

We have expanded this section of the methods now and include a table to more explicitly highlight what was done and which data was used.

In Figure 11, it would be worth stating what the assumed tectonic setting is. These are classic lithosphere-dimensioned yield envelopes for carbonate deformation including frictional and plastic flow laws for an assumed, fixed strain rate, geothermal gradient and tectonic stress state (compressional, extensional, strike-slip/transform boundaries). Please include the assumed tectonic setting in the caption. Similarly, please include the assumed strain rate in figures 12b and 12c.

We have added a brittle envelope for the thrust and normal fault setting alongside the Byerlee strength we had previously included in figure 11. Figure 12 has the strain rates included in the figure but we now refer to them in the caption more clearly.

I find the comparisons of flow laws for non-zero-XMgCO3 calcite aggregates and dolomite very interesting, and I wonder if the authors might add some possible explanations of the differences in the discussion of this manuscript. At the outset, differences in crystal symmetry and structure of calcite and dolomite, and the different slip systems of these two minerals offer obvious reasons that their flow laws for coarse-grained aggregates associated with intracrystalline deformation cannot readily be combined into one simple relation. However, it is fair to wonder why this doesn't work well for grain-size-sensitive deformation and processes at grain boundaries (grain boundary diffusion and sliding). We know little about the atomic structures of grain boundaries, so a relationship that includes grain size and XMgCO3 might be expected to work for inter-crystalline creep mechanisms in relatively "unstructured" carbonate grain boundaries. It could be that mean jump distances or frequencies might differ for dolomite grain boundaries (where dolomite grains are ordered in Ca and Mg) from those at calcite grain boundaries. This might mean that grain boundary diffusion depends more strongly on XMgCO3 than captured by grain-size-sensitive

calcite deformation results. Alternatively, nucleation (or reaction to add to one of the dolomite grains adjacent to a grain boundary) during diffusion creep and grain boundary sliding may differ at XMgCO3 values near 0.5. I don't think these potential differences can be evaluated without further data or information, but they are interesting possibilities. I wonder if grain-size-sensitive deformation of magnesite (at XMgCO3 =1) with the same mineral structure as calcite might fit the universal carbonate deformation law presented here better than flow laws reported for dolomite.

We have changed sections 5.2 and 5.3 accordingly. We now include magnesite data and discuss the effects that the reviewer raises.

Throughout, this manuscript is well written and I have only very minor editorial suggestions for rewording, which I will list below by line number:

Line 2 (abstract) – "… limestones and marbles. Such an"
The text has been changed accordingly

Line 100 – Was this normalization done just for calcite samples? Or both calcite and dolomite samples? Please specify explicitly ("… of natural calcite samples… " or "… of natural calcite and synthetic (?) dolomite samples …")
Line 108 – again please specify if just calcite samples or both calcite and dolomite. The max XMgCO3 value of 0.17 certainly suggests that dolomite data were not included in the global fit.
The text has been changed accordingly to specify and clarify the data used.

Line 115 – "… sample data are transformed to:"
The text has been changed accordingly

Line 117 – "and results for coarse grained …"
The text has been changed accordingly

Line 120 – "… exponents to collapse data to a one-to-one curve …"
The text was meant to read "on to one curve", thank you for drawing our attention to it: it has been changed accordingly

Line 128 – "… regimes that we think contribute to deformation and can be used in investigation and normalization. In our data there are two"
We have changed the text to:
"…regimes that we think contribute to deformation and then invert the normalisation for flow law parameters. In our data there are two …"

Line 129 – "… different deformation mechanisms are apparent: a low stress domain in which data have a slope of 1 in"
The text has been changed accordingly

Line 130 – "… a high stress domain characterized by a slope of 7 …"
The text has been changed accordingly

Line 147 – delete "then"
The text has been changed accordingly

Line 155 – suggest deleting "that"
The text has been changed accordingly

Line 156 – "… mechanisms; (2) XMgCO3 has two opposing effects …"
The text has been changed accordingly

Line 157 – "… coarse grained natural samples is only half that of the compositional sensitivity of fine grained"
The text has been changed accordingly

Line 160 – suggest replacing "visualized" by "illustrated"
The text has been changed accordingly

Line 171 – "... and coarse grained samples in sections 5.2 and 5.3."
The text has been changed accordingly

Line 174 – "The effect of magnesium on diffusion creep was first noted in the original work …"
The text has been changed accordingly

Line 177 – "... magnesium can be separated empirically from grain size. Therefore,"
The text has been changed accordingly

Lines 184-185 – "the observation of the same or broadly similar activation energies across Mg-carbonates (Herwegh et al., 2003). This observation suggests that mechanisms activated are thermodynamically equivalent, possibly the exact same mechanism for all chemistries."
We have changed the text to:
"…the observation of the same or broadly similar activation energies across Mg-carbonates (Herwegh et al., 2003), and more broadly across the whole Mg-Ca carbonate series (Holyoke et al., 2014). These observation suggests that mechanisms activated are thermodynamically equivalent, possibly the exact same mechanism for all chemistries."

Line 189 – "2008)) as reported for cation species self diffusion or diffusion creep of calcite (Herwegh …"
The text has been changed accordingly

Line 190 – "... one should expect that local spatial gradient are reduced as XMgCO3 increases, and magnesium"
The text has been changed accordingly

Line 191 – "... This is what we observe. ..."
The text has been changed accordingly

Line 192 – delete "which are derived from linearly spaced XMgCO3 values"
The text has been changed accordingly

Lines 196-197 – "... (fig. 4a and c). Indeed, our results suggest that this compositional effect on flow law extended beyond synthetic Mg-carbonates …"
We have changed the text to:
"Indeed, our results suggest that this compositional effect on creep extends beyond synthetic Mg-carbonates to natural calcite samples."

Line 198 – "strengthening by XMgCO3 may be due to solute-drag …"
The text has been changed accordingly

Line 199 – "to the progress of linear defects. This second possibility is not expected if chemical diffusion is substantial as discussed earlier"
We have changed the text to:
"This second possibility may be unlikely if chemical diffusion is substantial, as discussed earlier…"

Line 235 – delete "the framing of"
We have changed the text to:
"Reinforcing the framing of our discussion into fine and coarse grained samples, the rheology of the coarser grained starting material cannot…"

Line 238 – "0.3). Notably the newly defined XMgCO3-sensitive diffusion creep law …"
The text has been changed accordingly

Lines 245-246 – "... Lochseiten mylonite data. It is important to ask how well the equations fit carbonates …"
The text has been changed accordingly

Line 248 – "Equations 5-6 were compared …"
The text has been changed accordingly

Lines 251-253 – "... data appear to be fit well. In the case of diffusion creep of the Lochseiten mylonite, the data fit equation 5 at higher temperatures but not at lower temperatures (fig. 8c). Figure 8d shows that, as the second phase content rises to 10% and above, the observed rheologies depart from our model."
The text has been changed accordingly

Line 255 – "The normalization of experimental results of coarse-grained natural sample results was limited to …"
The text has been changed accordingly

Line 258 – "When compared to other coarse grained calcite experiments, the data fit the model relatively well (9a-c). However, they do"
We have changed the text to:
"Despite this, when compared to other coarse grained calcite experiments the data fit the model relatively well (fig. 9a-c) , but could not account for experiments run on dolomite or magnesite (fig. 9d).

Line 261 – "the spread of data points. …."
The text has been changed accordingly

Line 262 – delete "will"
The text has been changed accordingly

Line 263 – "there is a marked increase in strain. 'the fit clearly overestimates ..."
We choose to retain our formulation.

Line 264 – "... with in the range of Carrara marble experimental results. Regardless"
The text has been changed accordingly

Line 269 – "... result of differences in sample ..."
We choose to retain our formulation.

Line 271 – "... synthetic and natural for XMgCO3 values that"
We have changed the text to:
"Figure 10 shows data from fine and coarse grained samples, that are both synthetic and natural, with XMgCO3 values that are close together."

Line 272 – "... Data plotted for SMgCO3 = 0.07 come from a fine grained ..."
The text has been changed accordingly

Line 275 – "While data plotted for XMgCO3 = 0.03 comes from a fine grained ...
The text has been changed accordingly

Line 277 – "... is not restricted to synthetic samples alone – it is not an artifact"
The text has been changed accordingly

Lines 278-280 – "... there is a transition to a grain size sensitive dislocation creep field. We conclude from this that grain size sensitive and grain size insensitive dislocation creep fields are real."
The text has been changed accordingly

Line 292 – "a modified Hall-Petch relation has been proposed; the smaller the grain size, the stronger …"

The text has been changed accordingly
Lines 294-296 – "in the dislocation field appears to be similar to the d-3 dependence of diffusion creep. Moreover, figure 10 implies that there is a transition as grain size increases to a grain size insensitive rheology."
We choose to retain our formulation.

Line 298 – "To explain the ever diminishing ..."
The text has been changed accordingly

Line 300 – "... could transition from grain size strengthening to weakening and …"
The text has been changed accordingly

Lines 301-302 – "deformation as grain size increased. In this model the grain size sensitivity of dislocation accommodated deformation changes according to the storage and recovery of dislocations, and the dependence of these processes on grain size. Breithaupt et al. (2023)"
We choose to retain our formulation.

Lines 305-309 – "... Breithaupt, 2022). Thus, the apparent results for diffusion creep and grain size sensitive deformation of polycrystalline calcite might be linked to the storage and recovery in grains, and how these change with size of grains. The model of Breithaupt et al. (2023) also explains the variation (hardening and weakening) in the effect of grain size for different calcite experiments. …"
We choose to retain our formulation.

Lines 312- 313 – "et al., 2002). This is most likely for lower temperature experiments of Sly et al. (2019). While the grain sizes of Renner et al. (2002) and Herwegh et al. (2003) seem to cover a similar range …"
We choose to retain our formulation.

Lines 315-316 – "... vs area-weighted); grain size values of Renner et al. (2002) might be half those measured by Herwegh et al. (2003; cf. table 5 in Berhger et al., 2011)."
We choose to retain our formulation.

Line 319 – "... sensitivity is needed. Short of adapting"
The text has been changed accordingly

Lines 320-321 – "... the effect of XMgCO3, a pragmatic guide to modeling different rheologies based on grain size domains (Fig. 10)."
We choose to retain our formulation.

Line 323 – replace "if" by "for"
Line 325 – replace "if" by "for"
Line 326 – replace "if" by "for"
The text has been changed accordingly

Line 328 – "... 1987), figure 11 illustrates this approach …"
The text has been changed accordingly

Lines 329-331 – "it to a case for fine grained dislocation creep rheology that shows no grain size effect (dashed lines in fig. 11b), and the published flow laws for calcite (Schmid et al. 1980; Herwegh et al., 2003) fig. 11c). Regardless of which composite …"
The text has been changed accordingly

Lines 333-336 – "when modelling crustal strength. Figure 11 also shows that the fine grained dislocation creep rheology predicts low strengths in the crust with depth."
We have removed this section of text as it was not needed.

Line 338 – "Using temperature and grain size measurements for mylonites ..."
We have changed the text to:

"Using temperature and grain size measurements from natural carbonate shear zone mylonites we can account…"

We have changed the text to:
Figure 12 shows data from mylonites deformed at greenschist to amphibolite facies conditions from several thrusts of the Helvetic Nappes (Ebert et al., 2008) and from Naxos (Herwegh et al., 2011).

The text has been changed accordingly

The text has been changed accordingly

The text has been changed accordingly

We have changed the text to:
"Furthermore for a given molar fraction of magnesium, our review of the data show that at high stresses, at which dislocation mediated deformation will occur, finer grain sizes produce higher strain rates (fig. 10)."

The text has been changed accordingly

The text has been changed accordingly to the referees more measured phrasing, but we note for purposes of this review that questionable predictions lack usefulness.

The text has been changed accordingly

[revised manuscript text omitted]

---

## Author Response (AR2)

Dear Editors,

All technical corrections noted by the handling editor have been adopted in the manuscript.

Best wishes,

James Gilgannon